# Inconsistency-Aware Minimization: Improving Generalization with Unlabeled Data

## Abstract

Estimating the generalization gap and developing optimization methods that improve generalization are crucial for deep learning models, for both theoretical understanding and practical applications. Leveraging unlabeled data for these purposes offers significant advantages in real-world scenarios. This paper introduces a novel generalization measure, *local inconsistency*, derived from an information-geometric perspective on the parameter space of neural networks. A key feature of local inconsistency is that it can be computed without explicit labels. We establish theoretical underpinnings by connecting local inconsistency to Fisher information matrix and loss Hessian. Empirically, we demonstrate that local inconsistency correlates with the generalization gap. Based on these findings, we propose Inconsistency-Aware Minimization (IAM), which incorporates local inconsistency into the training objective. We demonstrate that in standard supervised learning settings, IAM enhances generalization, achieving performance comparable to that of existing methods such as Sharpness-Aware Minimization. Furthermore, IAM exhibits efficacy in semi- and self-supervised learning scenarios, where the local inconsistency is computed from unlabeled data.

## 1    Introduction

Estimating the generalization gap and optimizing models to perform well on unseen data are central challenges in deep learning. Prior work has linked the flatness of the loss landscape to generalization and proposed sharpness-driven optimizers; however, sharpness—often instantiated as the largest eigenvalue of the loss Hessian—does not by itself reliably predict the generalization gap across settings (Keskar et al., 2017; Dinh et al., 2017; Li et al., 2018; Garipov et al., 2018; Foret et al., 2021; Kwon et al., 2021; Kim et al., 2022; Zhuang et al., 2022; Andriushchenko et al., 2023).

Alternatively, some studies examine output-based measures such as *disagreement* (Jiang et al., 2022) and *inconsistency* (Johnson & Zhang, 2023), which can correlate with the generalization gap under certain conditions. However, because disagreement is non-differentiable, it is difficult to incorporate directly into the training process. Furthermore, inconsistency is impractical for training a single model, as it requires aggregating outputs form multiple models and data splits, which is a computationally expensive process.

In this work, we introduce *local inconsistency*, an information-geometric measure of output sensitivity in parameter space. Concretely, local inconsistency is defined as the worst-case (within an $\ell_2$ ball) KL divergence between the output distributions of a model and its perturbed counterpart. Crucially, it is (i) **computable from a single trained model** and (ii) **differentiable**, enabling both estimation and *direct regularization* within standard training pipelines. Furthermore, its computation (iii) **relies only on unlabeled data**, a key property that unlocks applications in label-constrained settings, including semi-/self-supervised learning.

We theoretically ground local inconsistency by connecting it to the Fisher information matrix (FIM) and the loss Hessian, showing that, under a local quadratic approximation, it is governed by the FIM's largest eigenvalue. This provides a complementary signal to traditional sharpness (e.g., $\lambda_{\max}(H)$), as we find that local inconsistency maintains a meaningful correlation with the generalization gap even in settings where sharpness measures falter.

Building on this, we propose *Inconsistency-Aware Minimization* (IAM), which incorporates local inconsistency into the training objective. IAM inherits the practical advantages of single-model training while uniquely enabling *regularization from unlabeled data*. On CIFAR-10/100 supervised benchmarks, IAM matches or surpasses sharpness-aware baselines. Crucially, its label-agnostic nature makes it a versatile regularizer for other learning paradigms; we show it boosts the performance of both the semi-supervised framework FixMatch and the self-supervised method SimCLR, demonstrating its broad applicability.

- **A computable and differentiable measure from unlabeled data.** We introduce *local inconsistency*, an information-geometric generalization measure that is *Model-intrinsic* and *label-free*, making it practical both to estimate and to *regularize* during training.

- **Theory: links to FIM/Hessian and to prior inconsistency.** We formalize connections from *local inconsistency* to the FIM (and via Gauss–Newton to the Hessian) and discuss an relationship to Johnson & Zhang (2023), clarifying how local inconsistency complements inconsistency while avoiding the multi-model costs of prior inconsistency measures.

- **Method: IAM for labeled, semi-/ self-supervised learning.** We develop IAM, which incorporate local inconsistency into the training objective. IAM achieves competitive or superior generalization to SAM in supervised tasks and, uniquely, *leverages unlabeled data* to improve semi- and self-supervised training.

## 2 RELATED WORK

Understanding and improving generalization in deep neural networks, especially given their large capacity and tendency to overfit (Zhang et al., 2017), remains a central challenge. While networks can memorize random labels (Zhang et al., 2017) and learn simple patterns before noise (Arpit et al., 2017), phenomena like double descent (Nakkiran et al., 2021) and the inadequacy of uniform convergence theory (Nagarajan & Kolter, 2019) highlight the need for novel generalization measures beyond loss-based metrics.

Traditional measures like VC-dimension often fall short. While spectrally-normalized margin bounds (Bartlett et al., 2017) and PAC-Bayes approaches offer insights, no single measure consistently predicts generalization (Jiang et al., 2019). Recently, disagreement (Jiang et al., 2022) and inconsistency (Johnson & Zhang, 2023) have shown promise, correlating well with the generalization gap, even when computed on unlabeled data. However, their reliance on training multiple models poses practical limitations for direct optimization in a single-model setup, underscoring the need for efficient, label-free, single-model generalization measures.

The geometry of the loss landscape, particularly the flatness of minima, has been extensively linked to generalization (Keskar et al., 2017; Li et al., 2018). However, the utility of sharpness as a sole predictor is debated due to issues like scale invariance (Dinh et al., 2017) and its correlation with training hyperparameters rather than true generalization (Andriushchenko et al., 2023). Indeed, some studies suggest that output inconsistency and instability can be more reliable predictors than sharpness (Johnson & Zhang, 2023). Information geometry has inspired reparametrization-invariant sharpness measure (Jang et al., 2022), but these can be computationally expensive. This context motivates our exploration of "local inconsistency", an alternative geometric measure focusing on output sensitivity within a parameter neighborhood, computable from unlabeled data using a single model.

Various regularization techniques, both explicit (e.g., dropout (Srivastava et al., 2014), batch normalization (Santurkar et al., 2018), Mixup (Zhang et al., 2018)) and implicit (e.g., SGD's bias (Hardt et al., 2016; Soudry et al., 2018)), aim to improve generalization. Methods like Sharpness-Aware Minimization (SAM, (Foret et al., 2021)) and ASAM (Kwon et al., 2021) directly optimize for flat minima and have shown significant improvements. Despite their success, the precise role of sharpness in generalization remains an active area of research (Jiang et al., 2019; Andriushchenko et al., 2023), further motivating the development of complementary approaches like our proposed IAM.

## 3 BACKGROUND AND PRELIMINARIES

In this section, we briefly review fundamental concepts and notations essential for understanding our proposed metric and its theoretical connections. We focus on probabilistic classification models, information geometry, and aspects of the loss landscape.

### 3.1 NOTATION AND PROBLEM SETUP

We consider probabilistic classification models. Let $x \in \mathcal{X}$ be a data point from the input space $\mathcal{X}$, and $y \in [C] = \{0, 1, \ldots, C-1\}$ be the corresponding class label, where $C$ is the total number of classes. The data pair $(x, y)$ are assumed to be drawn from an underlying distribution $\mathscr{D}$ over $\mathcal{X} \times [C]$. A model, parameterized by $\theta \in \mathbb{R}^m$, outputs a probability distribution over classes for a given input $x$. This is typically achieved by transforming a logit vector $z(x; \theta)$ through a softmax function: $f(x; \theta) = \mathrm{softmax}(z(x, \theta))$. Thus, $f(x; \theta) = [p(0|x; \theta), p(1|x; \theta), \ldots, p(C-1|x; \theta)]^{\top}$. Given a training dataset $Z_n = \{(x_i, y_i) : i = 1, \ldots, n\}$ drawn i.i.d. from $\mathscr{D}$, the model is typically trained by minimizing a loss function. For classification, the empirical Cross-Entropy (CE) loss will be written as $L(\theta) = \frac{1}{n} \sum_{i=1}^{n} l_i(\theta)$, where per-sample loss is $l_i(\theta) = l(x_i, y_i; \theta) = -\log p(y_i|x_i; \theta)$.

### 3.2 FISHER INFORMATION MATRIX (FIM) AND KL DIVERGENCE

The Fisher information matrix (FIM), $F(\theta)$, for the family of probability density $p(x, y; \theta) = p(x)p(y|x; \theta)$ parameterized by a parameters $\theta$ is defined as

$$
\begin{aligned}
F(\theta) &= \mathbb{E}_{x \sim p(x)} \left[ \mathbb{E}_{y \sim p(y|x;\theta)} \left[ \nabla_\theta l(x, y; \theta) \nabla_\theta l(x, y; \theta)^{\top} \right] \right] \\
&= \mathbb{E}_{x \sim p(x)} \left[ \nabla_\theta z(x; \theta) \left( \mathrm{diag}(f(x; \theta)) - f(x; \theta)f(x; \theta)^{\top} \right) \nabla_\theta z(x; \theta)^{\top} \right].
\end{aligned}
\tag{1}
$$

In practice, the expectation $\mathbb{E}_{p(x)}$ is often approximated by an empirical average over the available data (e.g., training data $\{x_i\}_{i=1}^{n}$ or unlabeled data).

The Kullback-Leibler (KL) divergence between the output distributions of a model with parameters $\theta$ and a slightly perturbed model $\theta + \delta$, $f(x; \theta)$ and $f(x; \theta + \delta)$, respectively, can be locally approximated using a second-order Taylor expansion with respect to $\delta$ as:

$$
\mathbb{E}_{x \sim p(x)} \left[ \mathrm{KL} \left( f(x; \theta) \| f(x; \theta + \delta) \right) \right] = \frac{1}{2} \delta^{\top} F(\theta) \delta + O(\|\delta\|_2^3).
\tag{2}
$$

### 3.3 LOSS HESSIAN AND GAUSS-NEWTON APPROXIMATION

The geometry of the empirical loss surface $L(\theta)$ is described by its Hessian matrix $H(\theta) = \nabla_\theta^2 L(\theta)$. For the CE loss, the Hessian can be approximated by the Gauss-Newton (GN) matrix, $G(\theta)$. The second of the per-sample CE loss $\ell_i(\theta)$ with respect to the logits $z_i = z(x_i; \theta)$, $\nabla_z^2 \ell_i(\theta) = \mathrm{diag}(f(x_i; \theta)) - f(x_i; \theta)f(x_i; \theta)^{\top}$, depends only on the model's output probabilities $f(x_i, \theta)$. Consequently, the per-sample GN term, $G_i(\theta) = \nabla_\theta z_i^{\top} (\nabla_z^2 \ell_i) \nabla_\theta z_i$, is equivalent to the FIM contribution in Eq. (1). The empirical GN matrix, $G(\theta) = \frac{1}{n} \sum_{i=1}^{n} G_i(\theta)$, thus often termed the empirical FIM, provides a positive semi-definite approximation to $H(\theta)$:

$$
H(\theta) \approx G(\theta) = F(\theta)
$$

and is frequently used in optimization (Martens, 2020; Pascanu & Bengio, 2014).

## 4 ACCESSING GENERALIZATION GAP VIA LOCAL INCONSISTENCY

This section introduces our proposed measure, local inconsistency, designed to capture the generalization gap. We first define local inconsistency and elucidate its theoretical underpinnings by connecting it to the FIM and the loss Hessian. We then discuss its relationship with inconsistency (Johnson & Zhang, 2023). Finally, we present empirical results demonstrating the correlation between local inconsistency and the generalization gap, comparing it with other common measures.

## 4.1 LOCAL INCONSISTENCY, $S_\rho(\theta)$

We introduce local inconsistency, $S_\rho(\theta)$, defined as:

$$S_\rho(\theta) = \max_{\|\delta\| \le \rho} \mathbb{E}_{x \sim p(x)}[\mathrm{KL}(f(x;\theta)\|f(x;\theta+\delta))], \tag{3}$$

which represents the sensitivity of the model's output distribution $f(x;\theta)$ with respect to the worst perturbations $\delta$, within an Euclidean ball of radius $\rho$ around the parameter $\theta$. Intuitively, a high value of $S_\rho(\theta)$ indicates that the model's output distribution is highly sensitive to small perturbations in parameter space. This sensitivity suggests potential instability or uncertainty in the model's predictions associated with the vicinity of $\theta$.

**Practical Advantages of $S_\rho$**    Local inconsistency shares a practical advantage with sharpness-based measures (Keskar et al., 2017; Foret et al., 2021) in that it can be calculated using a **single** trained model. Furthermore, like disagreement (Jiang et al., 2022) and inconsistency (Johnson & Zhang, 2023), our metric can be estimated using only **unlabeled** data. A notable advantage over inconsistency and disagreement estimation is that evaluating $S_\rho$ does not require training multiple model instances derived from the same training procedure and is **directly regularizable**. This potentially makes $S_\rho$ more computationally efficient and practical to compute, especially when model training is resource-intensive.

## 4.2 CONNECTION TO FIM AND HESSIAN

The relationship between our metric $S_\rho$ and the Fisher Information Matrix (FIM) can be established by leveraging the local quadratic approximation of the KL divergence, as outlined in Section 3. With this quadratic approximation, we can approximate $S_\rho(\theta)$ with the maximum eigenvalue of FIM, scaled by $\rho^2/2$:

$$S_\rho(\theta) \approx \max_{\|\delta\| \le \rho} \frac{1}{2}\delta^\top F(\theta)\delta = \frac{1}{2}(\rho v_{\max})^\top F(\theta)(\rho v_{\max}) = \frac{1}{2}\rho^2 \lambda_{\max}, \tag{4}$$

where $v_{\max}$ is the eigenvector corresponding to the largest eigenvalue $\lambda_{\max}$ of $F(\theta)$. Remarkably, this approximation requires only the model $\theta$ and unlabeled data (used to compute the expectation).

The Fisher Information Matrix $F(\theta)$, to which $S_\rho(\theta)$ is related via its maximum eigenvalue, also connects to the Hessian of the loss function $H(\theta)$. As detailed in Section 3, for Negative Log Likelihood losses such as CE, the Hessian can be approximated by the Gauss-Newton matrix $G(\theta)$, equivalent to empirical FIM computed using training data.

Consequently, when calculating $S_\rho(\theta)$ using the training data, it approximates $\frac{1}{2}\rho^2 \lambda_{\max}(G(\theta))$. Given that $G(\theta)$ often provides a good approximation to the true loss Hessian near a local minimum, $S_\rho(\theta)$ therefore offers insights into the maximum curvature of the loss landscape in that vicinity.

## 4.3 LOCAL INCONSISTENCY AND GENERALIZATION BOUNDS (FIM FORM)

Under near interpolation, which is a standard regime in modern deep learning (Zhang et al., 2017), the empirical Hessian splits into a Fisher/Gauss–Newton term plus a small residual, which lets us replace $\lambda_{\max}(H_S(\theta))$ with $\lambda_{\max}(F_S(\theta))$ up to a spectral slack.

**Theorem 4.1** (FIM-based generalization bound)**.** *Under the same assumption of Theorem 3.1 of Luo et al. (2024), for any $\xi \in (0,1)$ and $\rho > 0$, with a probability over $1 - \xi$ over choice of $S \sim \mathscr{D}$, we have*

$$L_{\mathscr{D}}(\theta) \le L_S(\theta) + \frac{\rho^2}{2}\Big(\lambda_{\max}\big(F_S(\theta)\big) + \varepsilon_R\Big) + \frac{C\rho^3}{6} + (\textit{Complexity term}),$$

*where $n$ is the number of samples.*

In particular, at (near) interpolation ($\varepsilon_R \approx 0$), the Hessian term is replaced by $\lambda_{\max}(F_S(\theta))$ with no degradations. We defer the exact the complexity term and the proof to Appendix A.

This bound suggests that minimizing a combination of the empirical loss $L_S(\theta)$ and the local inconsistency $S_\rho(\theta)$ can lead to a lower upper bound on the true risk $L_D(\theta)$. This provides a theoretical motivation for our Inconsistency-Aware Minimization (IAM) framework, which aims to find solutions that are not only accurate on the training data but also exhibit low output sensitivity in the parameter space, as measured by $S_\rho(\theta)$.

### 4.4 RELATION WITH INCONSISTENCY IN JOHNSON & ZHANG (2023)

Local inconsistency exhibits an interesting relationship to the inconsistency in Johnson & Zhang (2023) defined as:

$$\mathcal{C}_P = \mathbb{E}_{Z_n}\mathbb{E}_{\theta,\theta'\sim\Theta_{P|Z_n}}\mathbb{E}_{x\sim p(x)}[\mathrm{KL}(f(x;\theta)\|f(x;\theta'))].$$

We consider the conditional inconsistency for a fixed $Z_n$, denoted $\mathcal{C}_{P|Z_n}$, without outer expectation. Then our proposed metric, $S_\rho(\theta_{Z_n})$, is approximately proportional to the conditional inconsistency $\mathcal{C}_{P|Z_n}$:

$$\frac{m}{2C}\mathcal{C}_{P|Z_n} \lesssim S_\rho(\theta_{Z_n}) \lesssim \frac{m}{2}\mathcal{C}_{P|Z_n}, \tag{5}$$

under certain assumptions, such as assuming the parameter posterior $\Theta_{P|Z_n}$ as a distribution with isotropic covariance and $\theta_{Z_n}$ as mean. This connection arises because both metrics are related to the local geometry captured by the FIM at $\theta_{Z_n}$, with $S_\rho$ being linked to its maximum eigenvalue and $\mathcal{C}_{P|Z_n}$ to its trace. Practically, the eigenspectra of the FIM of a neural network are observed to be dominated by a few large eigenvalues (specifically related to the number of classes, $C$ in classification task) while remaining eigenvalues are near zero (Sagun et al., 2018; Papyan, 2018; 2019; 2020; Karakida et al., 2019; 2021). This observation indicates that the ratio $\lambda_{\max}(F(\theta))/\mathrm{Tr}(F(\theta))$ is larger than $\frac{1}{C}$ ($C \ll m$). For detailed derivation, please see Appendix B.

### 4.5 ESTIMATING $S_\rho(\theta)$

Directly computing $S_\rho(\theta)$ requires solving the maximization problem over the high-dimensional parameter perturbation $\delta$. For deep neural networks, finding the exact maximum within the $L_2$-ball of radius $\rho$ is generally intractable. Therefore, we employ numerical approximation methods.

For small perturbations $\delta$, the expected KL divergence can be accurately approximated by a second-order Taylor expansion involving the Fisher Information Matrix (FIM), $F(\theta)$, as Eq. (2) in Section 3 . Under quadratic approximation, as discussed in Section 4.2, the optimal perturbation $\delta^* = \rho v_{\max}$, the maximum value is then $S_\rho(\theta) = \frac{1}{2}\rho^2\lambda_{\max}$, and the gradient of the approximated KL divergence with respect to $\delta$ is $F(\theta)\delta$.

This connection motivates not an usual Projected Gradient Ascent, that update $\delta_{k+1} \leftarrow \Pi_{\{\delta_k:\|\delta_k\|\le\rho\}}(\delta_k + \eta F(\theta)\delta_k)$, but an iterative gradient ascent approach that update

$$\delta_{k+1} = \frac{\rho}{\|F(\theta)\delta_k\|}F(\theta)\delta_k, \qquad \delta_0 = \varepsilon \sim \mathcal{N}\left(0, \frac{\sigma^2}{m}I_m\right),$$

where $\sigma^2$ is initial noise scale. Iterative gradient ascent is precisely one iteration of the Power Iteration method used to find the dominant eigenvector of $F(\theta)$.

#### 4.5.1 ALGORITHM FOR ESTIMATING $S_\rho(\theta)$

Based on the above, we propose Algorithm 1 to estimate $S_\rho(\theta)$. This algorithm performs $K$ steps of normalized gradient ascent (effectively, Power Iteration under the quadratic approximation) to find an approximate maximizing perturbation $\delta^*$. Algorithm 1 requires $K$ gradient computation. See Appendix E for detail practical consideration about Algorithm 1.

---

**Algorithm 1** Estimation of $S_\rho(\theta)$

---

1: **Input:** model parameter $\theta \in \mathbb{R}^m$, noise scale $\sigma^2$,
2: radius $\rho > 0$, number of steps $K \ge 1$
3: **Initialize** $\delta_0$ randomly with $\mathcal{N}(0, \frac{\sigma^2}{m}I_m)$
4: **for** $k = 0$ to $K - 1$ **do**
5:     Compute $g_k = \nabla_\delta\mathbb{E}_{x\sim p(x)}\mathrm{KL}(f(x;\theta)\|f(x;\theta+\delta))|_{\delta=\delta_k}$
6:     Update perturbation: $\delta_{k+1} = \rho\frac{g_k}{\|g_k\|_2}$
7: **end for**
8: **return** $\mathbb{E}_{x\sim p(x)}\mathrm{KL}(f(x;\theta)\|f(x;\theta+\delta_K))$

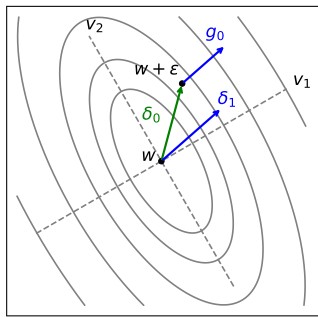

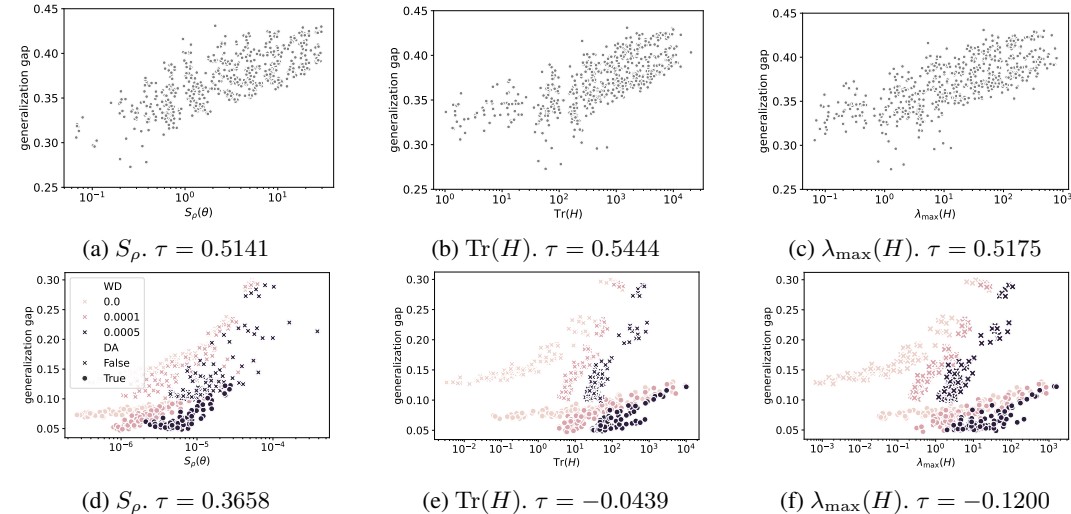

Figure 1: Local inconsistency and sharpness measures vs the generalization gap.

## 4.6 EMPIRICAL RESULTS

To assess the predictive capability of local inconsistency $S_\rho$ for the generalization gap, we conducted experiments on CIFAR-10. We trained two distinct architectures, a 6-layer CNN (6CNN) and a Wide Residual Network (WRN28-2) (Zagoruyko & Komodakis, 2017), under various hyperparameter settings (details in Appendix E). $S_\rho$ was estimated using a disjoint, unlabeled data set. For comparison, we also computed two common sharpness-based measures: the trace, $\mathrm{Tr}(H)$, and the maximum eigenvalue, $\lambda_{\max}(H)$.

Figure 1 presents scatter plots of these metrics against the generalization gap, with Kendall's Tau ($\tau$) reported for each. For the simpler 6CNN model (top row), $S_\rho$ ($\tau = 0.5141$) exhibited a positive correlation with the generalization gap, comparable to $\mathrm{Tr}(H)$ ($\tau = 0.5444$) and $\lambda_{\max}(H)$ ($\tau = 0.5175$). This suggests that for smaller models, various geometric measures may similarly capture aspects of generalization. However, for the larger WRN28-2 model with data augmentation (bottom row), a more nuanced behavior emerged. As noted by Andriushchenko et al. (2023), different training configurations can form distinct solution subgroups. In our WRN28-2 experiments, $\mathrm{Tr}(H)$ and $\lambda_{\max}(H)$ showed positive correlations only within such subgroups, but exhibited negative overall correlations globally ($\tau = -0.0439$ and $\tau = -0.1200$, respectively). In stark contrast, our $S_\rho$ maintained a positive, albeit reduced, correlation across all settings ($\tau = 0.3658$).

This divergence, particularly with larger models and data augmentation, suggests that local inconsistency captures information about the generalization gap that is distinct from, or complementary to, traditional Hessian-based sharpness. While the predictive utility of sharpness metrics can be confounded by these subgroup effects, $S_\rho$ demonstrates more consistent global predictiveness, hinting at its potential as a more robust generalization indicator in complex training scenarios.

## 5 INCONSISTENCY-AWARE MINIMIZATION (IAM): INCORPORATING LOCAL INCONSISTENCY INTO THE OBJECTIVE

Our empirical findings suggest that local inconsistency, $S_\rho(\theta)$ defined in Eq. (3), correlates with the generalization gap. This motivates its use as a regularizer to guide the optimization towards solutions that not only fit the training data, but also exhibit low sensitivity in their output distributions with respect to parameter perturbations. We propose two strategies to incorporate local inconsistency into the training objective.

---

**Algorithm 2** Inconsistency-Aware Minimization (IAM-S)

---

1: **Input:** Initial model parameters $\theta^0$; Learning rate $\eta$; neighborhood size $\rho$; training set $Z_n$; Batch size $b$; Number of steps $K$ for Algorithm 1.
2: **while** not converged **do**
3:      Sample batch $\{(x_i, y_i)\}_{i=1}^b$.
4:      Compute $\delta_K$ from Algorithm 1 using current $\theta$, $\rho$, and data $\{x_i\}_{i=1}^b$.
5:      Compute gradient $g = \nabla_\theta L(\theta)|_{\theta + \delta_K}$
6:      Update parameters: $\theta \leftarrow \theta - \eta g$.
7: **end while**
8: **Return** optimized parameters $\theta$.

---

1. **Direct Regularization (IAM-D)**: This approach directly penalizes local inconsistency by adding it to the standard training loss $L(\theta)$:

$$L_{\text{IAM-D}}(\theta) = L(\theta) + \beta S_\rho(\theta) = L(\theta) + \beta \max_{\|\delta\|_2 \leq \rho} \frac{1}{n} \sum_{i=1}^n \text{KL}(f(x_i, \theta) \| f(x_i, \theta + \delta)), \quad (6)$$

where $\beta > 0$ is a hyperparameter balancing the trade-off. This objective seeks parameter values $\theta$ for which the model outputs are consistent across the neighborhood defined by $\rho$.

2. **SAM-like Approach (IAM-S)**: Inspired by SAM (Foret et al., 2021), this method aims to find parameters $\theta$ that reside in a neighborhood of uniformly low loss by minimizing the loss at an adversarially perturbed point $\theta + \delta^*$:

$$L_{\text{IAM-S}}(\theta) = L(\theta + \delta^*), \quad \text{where } \delta^* = \arg\max_{\|\delta\|_2 \leq \rho} \frac{1}{n} \sum_{i=1}^n \text{KL}(f(x_i, \theta) \| f(x_i, \theta + \delta)). \quad (7)$$

Here, $\delta^*$ is the perturbation that maximizes the local inconsistency term. Note that the objective minimizes the original loss $L$ at the perturbed point $\theta + \delta$:

$$L(\theta + \delta) \approx L(\theta) + \delta^\top \nabla_\theta L(\theta) + \frac{1}{2} \delta^\top G(\theta) \delta.$$

Thus, IAM-S implicitly minimizes the principal eigenvalues of $G(\theta)$, equivalent to empirical FIM.

### 5.1 ALGORITHM FOR IAM-D AND IAM-S

Optimizing $L_{\text{IAM-D}}(\theta)$ and $L_{\text{IAM-S}}(\theta)$ involves a min-max procedure. The inner maximization to find $\delta^*$ (i.e., computing $S_\rho(\theta)$ and the corresponding $\delta^*$) is performed using an Algorithm 1 with current mini-batch, typically for $K = 1$ to match the number of additional gradient computations to that of SAM. We discuss effectiveness of $\delta_1$ with intuitive example in Appendix C. Moreover, as $K$ increases, IAM benefits from a more accurate estimation of local inconsistency, offering a trade-off between performance and cost (see Appendix D.1.1).

IAM-D simply add the $\beta S_\rho(\theta)$ with $\delta_K$ to the $L(\theta)$, and then update $\theta$ with standard SGD. The outer minimization step of IAM-S updates $\theta$ based on the gradient of the loss $L(\theta + \delta_K)$ dropping the second-order terms same with SAM: $\nabla_\theta L_{\text{IAM-S}}(\theta) \approx \nabla_\theta L(\theta)|_{\theta = \theta + \delta_K}$ as summarized in Algorithm 2.

### 5.2 EMPIRICAL EVALUATION IN SUPERVISED LEARNING

We evaluated the performance of IAM against SGD, SAM, and ASAM (Kwon et al., 2021) in image classification tasks. WRN (Zagoruyko & Komodakis, 2017) served as the baseline model, trained on CIFAR-{10, 100}, F-MNIST, and SVHN with basic augmentations. We used WRN-16-8 for CIFAR-{10, 100}, and WRN-28-10 for F-MNIST and SVHN. Optimal hyperparameters (determined via a grid search) for IAM-D were found to be $\beta = 1.0$, $\rho = 0.1$ for CIFAR-10, and $\beta = 10.0$, $\rho = 0.1$ for CIFAR-100, and for IAM-S were $\rho = 0.1, 0.5$ in CIFAR-10 and CIFAR-100 respectively. Table 1 summarizes the test error rates. Both IAM-D and IAM-S variants not only reduce test error compared to SGD but also achieve performance comparable to SAM and ASAM. In particular, on CIFAR-100, IAM-S outperforms SAM by a margin of 0.75%, demonstrating its effectiveness in more complex datasets. See Appendix D.3 for finetuning results of ViT (Dosovitskiy et al., 2021) on CIFAR.

Table 1: Test Error (mean $\pm$ stderr) of SGD, SAM, ASAM, and IAM across datasets.

| Dataset | SGD | SAM | ASAM | IAM-D | IAM-S |
|---|---|---|---|---|---|
| CIFAR-10 | 3.68 $\pm$0.04 | 3.31 $\pm$0.01 | **3.15** $\pm$0.02 | 3.28 $\pm$0.06 | 3.28 $\pm$0.03 |
| CIFAR-100 | 19.17 $\pm$0.19 | 17.63 $\pm$0.12 | 17.15 $\pm$0.11 | 17.16 $\pm$0.03 | **16.82** $\pm$0.01 |
| F-MNIST | 4.45 $\pm$0.05 | 4.13 $\pm$0.02 | 4.11 $\pm$<0.01 | 4.13 $\pm$0.04 | **4.10** $\pm$0.05 |
| SVHN | 3.82 $\pm$0.06 | 3.47 $\pm$0.09 | 3.24 $\pm$0.04 | **3.13** $\pm$0.06 | 3.13 $\pm$0.01 |

Figure 2 illustrates the evolution of local inconsistency $S_\rho(\theta)$ and test accuracy during training for SGD and IAM-D. IAM-D effectively suppresses the increase in $S_\rho(\theta)$ and mitigates overfitting, particularly evident after learning rate decay points where test accuracy for SGD can degrade. Both on CIFAR-10, 100 (Figure 2), IAM-D maintains $S_\rho(\theta)$ below SGD. Although second LR decay temporarily reduces inconsistency for both, SGD's inconsistency quickly rebounds, unlike the stable behavior of IAM-D. These observations suggest that minimizing local inconsistency helps confine the model to parameter regions with smoother output distributions, correlating with the generalization improvements shown in Table 1.

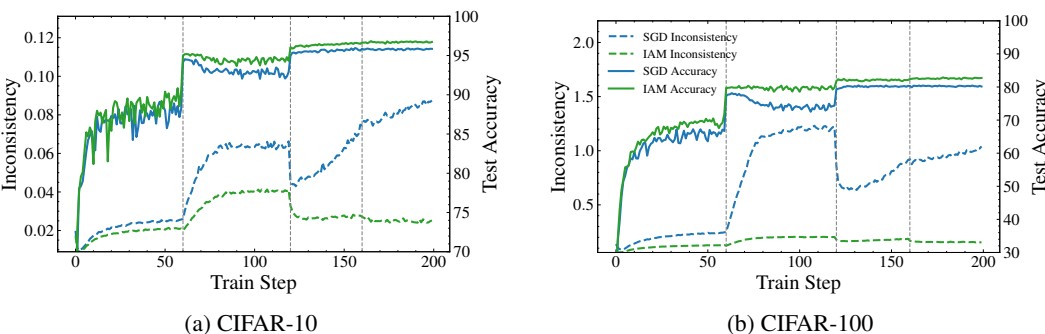

(a) CIFAR-10                                              (b) CIFAR-100

Figure 2: The evolution of the local inconsistency $S_\rho(\theta)$ and test accuracy with SGD and IAM-D.

To verify the scalability of our approach, we extend our evaluation to the large-scale ImageNet dataset using a ResNet-50 architecture. Following the experimental setting of Foret et al. (2021), we train the model with a batch size of 1024, initial learning rate of 0.1, and basic augmentations. We set $\rho = 0.2$ for IAM-S and $\rho = 0.05$ for SAM. As shown in Table 2, IAM-S outperforms the standard SGD baseline in both Top-1 and Top-5 error. We report the best score achieved by each SGD either 200 epochs or the 400 epochs. IAM-S trained for only 200 epochs achieves a lower error rate (21.72%) than SGD trained for 400 epochs (22.80%), demonstrating improved generalization. See Table 5 in Appendix D.2 for the results of each epochs.

Table 2: Top-1 and Top-5 error (mean $\pm$ stderr) of ResNet-50 trained 200 epochs on ImageNet.

| | SGD | SAM | IAM-S |
|---|---|---|---|
| Top-1 | 22.66 $\pm$ 0.12 | 21.80 $\pm$ 0.12 | **21.72** $\pm$ 0.07 |
| Top-5 | 6.51 $\pm$ 0.06 | 5.99 $\pm$ 0.04 | **5.90** $\pm$ 0.02 |

While SAM relies on the gradient of the training loss to determine the perturbation direction, our proposed IAM derives perturbations via the gradient of the KL divergence. The generalization performance of IAM-S empirically demonstrates that this single-step perturbation, $\delta_1 \approx F(\theta)\epsilon$, is not merely stochastic; rather, it effectively captures the principal eigenspace of the FIM. This suggests that even a minimal computational effort ($K = 1$) is sufficient to identify a meaningful direction, distinguishing it from random perturbations.

## 5.3 IAM FOR LEARNING WITH LIMITED OR NO EXPLICIT LABELS

A key advantage of local inconsistency is its computability from unlabeled data, making IAM well-suited for scenarios with limited or no explicit supervision. We demonstrate this in semi-supervised and self-supervised learning settings. IAM-D can be seamlessly "plugged in" to complex pipelines like FixMatch (Sohn et al., 2020) or SimCLR (Chen et al., 2020) by adding penalty term $\beta S_\rho(\theta)$ to the original objective. Detailed experimental settings are listed in Appendix E.

Table 3: Test error (mean $\pm$ stderr) with semi-supervised setting on CIFAR-10 and CIFAR-100

|  | CIFAR-10 | | CIFAR-100 | |
| --- | --- | --- | --- | --- |
|  | 250 labels | 4000 labels | 2500 labels | 10000 labels |
| SGD | $63.82 \pm 0.18$ | $22.45 \pm 0.40$ | $68.91 \pm 0.43$ | $45.94 \pm 0.35$ |
| SAM | $63.91 \pm 0.18$ | $19.95 \pm 0.22$ | $69.53 \pm 0.79$ | $43.30 \pm 0.11$ |
| IAM-D | $\mathbf{61.77} \pm 0.09$ | $\mathbf{15.07} \pm 0.14$ | $\mathbf{66.98} \pm 0.01$ | $\mathbf{40.02} \pm 0.13$ |
| FixMatch | $6.26 \pm 0.39$ | $4.10 \pm 0.17$ | $32.84 \pm 0.40$ | $22.93 \pm 0.05$ |
| FixMatch + IAM-D | $\mathbf{5.30} \pm 0.08$ | $\mathbf{3.88} \pm 0.02$ | $\mathbf{28.95} \pm 0.59$ | $\mathbf{21.99} \pm 0.04$ |

**Semi-Supervised Learning.** We demonstrate the advantage of IAM in a label-scarce setting on CIFAR-{10, 100}. Our method, IAM-D, optimizes a joint objective: the standard cross-entropy loss on the labeled subset, plus the local inconsistency penalty computed over the entire mini-batch (both labeled and unlabeled samples). The results in Table 3 show that IAM-D consistently outperforms both SGD and SAM. Furthermore, to highlight its versatility, we integrated IAM-D into the strong FixMatch framework (Sohn et al., 2020). This combination significantly lowers the test error both on CIFAR-10 and CIFAR-100, demonstrating that IAM-D can serve as an effective plug-and-play regularizer to enhance state-of-the-art Semi-supervised learning methods.

This approach contrasts with methods like SAM, which can only promote flatness over the small, labeled subset. Simply applying SAM to labeled loss of FixMatch fails to improve generalization (see Appendix D.4). A critical insight is that flatness measured on a sparse set of labeled points may not reflect true flatness across the entire data distribution. By leveraging second-order information from abundant unlabeled data, IAM-D seeks a more generalizable minimum.

**Self-Supervised Learning (SSL).** The label-agnostic nature of IAM makes it directly applicable to SSL objectives. We integrated IAM-D into the SimCLR framework (Chen et al., 2020), training a ResNet-18 (He et al., 2015) encoder on CIFAR-10. Performance was evaluated using linear probing. The local inconsistency term for IAM-D was computed using the model's projection-head outputs. Figure 3 shows that SimCLR trained with IAM-D (SimCLR-IAM) achieves higher test accuracy on the downstream linear classification task compared to vanilla SimCLR (SimCLR-SGD). Furthermore, SimCLR-IAM tends to converge faster in terms of test error and also minimizes the SimCLR training loss more rapidly, despite the additional local inconsistency regularization. This suggests that controlling local inconsistency is beneficial even when no explicit labels are available during representation learning.

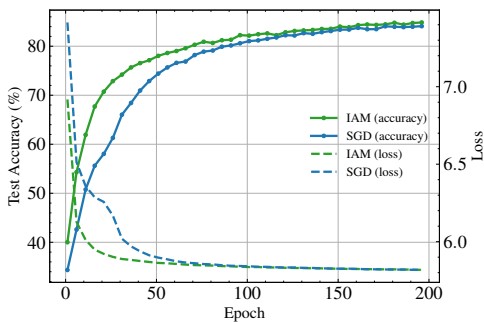

Figure 3: Test accuracy on linear probe and SimCLR training loss for ResNet-18 on CIFAR-10, comparing SimCLR trained with SGD (SimCLR-SGD) versus SimCLR with IAM-D (SimCLR-IAM).

# 6 CONCLUSION

In this work, we introduced "local inconsistency," a novel information-geometric generalization measure computable from a single model using only unlabeled data. We theoretically linked it to the Fisher Information Matrix (FIM) and the loss Hessian. Empirically, local inconsistency correlates with the generalization gap and exhibits distinct characteristics from traditional sharpness-based metrics.

Based on this, we proposed Inconsistency-Aware Minimization (IAM), an optimization framework that directly incorporates local inconsistency into the training objective. IAM enhances generalization in supervised learning, matching or exceeding that of Sharpness-Aware Minimization (SAM). Crucially, IAM proves effective in semi- and self-supervised learning by leveraging unlabeled data for local inconsistency computation, improving performance in label-scarce settings.

These findings offer a practical and theoretically-grounded approach to improving model generalization, particularly valuable in real-world applications where labeled data is limited. Future research could focus on exploring the scalability and applicability of IAM to a wider array of modern model architectures and other tasks or on developing computationally efficient version of IAM.

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

# A  PROOF OF THE FIM-BASED GENERALIZATION BOUND

We provide a self-contained derivation of the FIM-form bound stated in Theorem 4.1. Throughout, let $L_S(\theta) = \frac{1}{n} \sum_{i=1}^{n} \ell(f(x_i; \theta), y_i)$ be the empirical cross-entropy with logits $z(x; \theta) \in \mathbb{R}^C$, probabilities $f(x; \theta) = \text{softmax}(z)$, and $J(x; \theta) := \nabla_\theta z(x; \theta) \in \mathbb{R}^{C \times d}$. We write $H_S(\theta) := \nabla^2 L_S(\theta)$ and define the empirical Fisher

$$F_S(\theta) := \frac{1}{n} \sum_{i=1}^{n} J_i^\top \big( \text{diag}(f(x_i; \theta)) - f(x_i; \theta) f(x_i; \theta)^\top \big) J_i,$$

where $J_i := J(x_i; \theta)$.

**Assumption (near interpolation).**  There exists $\varepsilon_R \geq 0$ such that the residual

$$R_S(\theta) := \frac{1}{n} \sum_{i=1}^{n} \sum_{k=1}^{C} (f(x_i; \theta) - y_i)_k \, \nabla_\theta^2 z_k(x_i; \theta) \quad \text{satisfies} \quad \|R_S(\theta)\|_2 \leq \varepsilon_R. \qquad \text{(A1)}$$

STEP 1: HESSIAN–FIM DECOMPOSITION FOR SOFTMAX–CE

**Lemma A.1** (Gauss–Newton (=FIM) + residual).  *For each sample $i$, with loss $\ell_i := \ell(f(x_i; \theta), y_i)$,*

$$\nabla_\theta \ell_i = J_i^\top (f(x_i; \theta) - y_i)$$

$$\nabla_\theta^2 \ell_i = J_i^\top \big( \text{diag}(f(x_i; \theta)) - f(x_i; \theta) f(x_i; \theta)^\top \big) J_i + \sum_{k=1}^{C} (f(x_i; \theta) - y_i)_k \, \nabla_\theta^2 z_k(x_i; \theta).$$

*Averaging over $i$ yields $H_S(\theta) = F_S(\theta) + R_S(\theta)$.*

*Proof.* Since $\ell(p, y) = -\sum_k y_k \log p_k$ and $p = \text{softmax}(z)$, $\frac{\partial \ell}{\partial z} = p - y$. By the chain rule, $\nabla_\theta \ell_i = J_i^\top (f(x_i; \theta) - y_i)$. Differentiating once more,

$$\nabla_\theta^2 \ell_i = J_i^\top \Big( \frac{\partial f(x_i; \theta)}{\partial z_i} \Big) J_i + \sum_{k=1}^{C} \Big( \frac{\partial \ell_i}{\partial z_{ik}} \Big) \nabla_\theta^2 z_k(x_i, \theta),$$

and $\frac{\partial f(x_i; \theta)}{\partial z_i} = \text{diag}(f(x_i; \theta)) - f(x_i; \theta) f(x_i; \theta)^\top$ for softmax. Using $\frac{\partial \ell_i}{\partial z_{ik}} = (f(x_i; \theta) - y_i)_k$ gives the stated identity. Averaging over $i$ completes the proof. □

STEP 2: SPECTRAL CONTROL VIA WEYL'S INEQUALITY

**Lemma A.2** (Hessian vs. FIM eigenvalues).  *If $H_S = F_S + R_S$ with $F_S, R_S$ symmetric, then*

$$\lambda_{\max}(H_S) \leq \lambda_{\max}(F_S) + \|R_S\|_2.$$

Combining Lemma A.1 with Assumption equation A1 and Lemma A.2 gives

$$\lambda_{\max}\big( H_S(\theta) \big) \leq \lambda_{\max}\big( F_S(\theta) \big) + \varepsilon_R. \qquad (8)$$

STEP 3: FROM THE HESSIAN-BASED BOUND TO THE FIM FORM

We recall the Hessian-based bound of Luo et al. (2024) (Theorem 3.1) under the assumption that the loss function is bounded by $L$, the third-order partial derivative of the loss function is bounded by $C$, and $L_\mathscr{D}(\theta) \leq \mathbb{E}_{\varepsilon \sim \mathcal{N}(0, \sigma^2 I_m)} L_\mathscr{D}(\theta + \varepsilon)$.

$$L_\mathscr{D}(\theta) \leq L_S(\theta) + \frac{m \sigma^2}{2} \lambda_{\max}\big( H_S(\theta) \big) + \frac{C m^3 \sigma^3}{6} \qquad (9)$$

$$+ \frac{L}{2\sqrt{n}} \sqrt{m \log\big(1 + \frac{\|\theta\|^2}{\rho^2}\big) + 2 \log \frac{1}{\xi} + 4 \log(n + m) + O(1)}. \qquad (10)$$

**Theorem A.3** (FIM-based generalization bound; Theorem. 4.1). *Assume that the loss function is bounded by $L$, the third-order partial derivative of the loss function is bounded by $C$, and $L_{\mathscr{D}}(\theta) \le \mathbb{E}_{\varepsilon \sim \mathcal{N}(0, \sigma^2 I_m)} L_{\mathscr{D}}(\theta + \varepsilon)$. For any $\xi \in (0, 1)$ and $\rho > 0$, with a probability over $1 - \xi$ over choice of $S \sim \mathscr{D}$, we have*

$$
L_{\mathscr{D}}(\theta) \ \le \ L_S(\theta) + \frac{\rho^2}{2}\Big(\lambda_{\max}\big(F_S(\theta)\big) + \varepsilon_R\Big) + \frac{C\rho^3}{6}
$$

$$
+ \frac{L}{2\sqrt{n}}\sqrt{m \log\big(1 + \frac{\|\theta\|^2}{\rho^2}\big) + 2\log\frac{1}{\xi} + 4\log(n + m) + O(1)}, \tag{11}
$$

*where $n$ is the number of samples and $\rho = \sqrt{m}\sigma$.*

In particular, at (near) interpolation where $\varepsilon_R \approx 0$ ($L_S(\theta) \approx 0$), the Hessian term is replaced by $\lambda_{\max}(F_S(\theta))$ without degradation.

*Proof.* Substitute equation 8 into equation 9. □

# B   RELATION BETWEEN OUR METRIC AND INCONSISTENCY

This section outlines an approximate derivation relating the model output inconsistency $\mathcal{C}_P$, as defined by Johnson & Zhang (2023), to the local sensitivity metric $S_\rho(w)$ defined previously. we will show simple demonstrations that these two metrics are related primarily through the Fisher Information Matrix (FIM), under specific assumptions like isotropic covariance. Then will show results with anisotropic covariance.

**Definitions**

- **Inconsistency** ($\mathcal{C}_P$): Measures the average difference (in terms of KL divergence) between the outputs of models generated by a stochastic training procedure $P$ applied to the same training data $Z_n$. The average is taken over draws of the training data $Z_n$ and pairs of models $(\Theta, \Theta')$ drawn from the conditional distribution $\Theta_{P|Z_n}$.

$$
\mathcal{C}_P = \mathbb{E}_{Z_n}\mathbb{E}_{\Theta, \Theta' \sim \Theta_{P|Z_n}}\mathbb{E}_X[\mathrm{KL}(f(\Theta, X)\|f(\Theta', X))]
$$

  Here, $\Theta_{P|Z_n}$ denotes the distribution over parameters resulting from applying procedure $P$ to dataset $Z_n$.

- **Local Sensitivity** ($S_\rho(w)$): Measures the expected maximum change in the model's output distribution within a $\rho$-radius ball around a specific parameter vector $w$. For consistency with the derivation below, we use the form where the expectation is inside the maximization.

$$
S_\rho(\theta) = \max_{\|\delta\|_2 \le \rho} \mathbb{E}_x[\mathrm{KL}(f(x, \theta)\|f(x, \theta + \delta))]
$$

  Here, $\delta \in \mathbb{R}^d$ is a perturbation to the parameters $w$.

**Assumptions**   The following derivation relies on several key assumptions:

1. **Isotropic Covariance Posterior Assumption**: For a given training set $Z_n$, the conditional parameter distribution $\Theta_{P|Z_n}$ can be approximated by an isotropic distribution centered at a specific parameter vector $w_{Z_n}$ derived from $Z_n$: $\mathbb{E}[\Theta_{P|Z_n}] = w_{Z_n}$, $\mathrm{Cov}[\Theta_{P|Z_n}] = s^2\mathbf{I}_d$, where $s^2$ is a small variance. This approximation is motivated by studies interpreting Stochastic Gradient Descent (SGD) as a form of approximate Bayesian inference, where the distribution of parameters after training can resemble a Gaussian centered near a mode of a posterior distribution related to the loss function Mandt et al. (2018).

2. **Validity of Second-Order KL Approximation**: The KL divergence between outputs of models with slightly different parameters can be accurately approximated by a quadratic form involving the Fisher Information Matrix (FIM). This relies on the parameter difference being small, implying $s^2$ must be small.

3. **Effective FIM Constancy in Expectation**: The variations of the FIM $F(\Theta')$ for $\Theta' \sim \mathcal{N}(w_{Z_n}, s^2\mathbf{I}_d)$ around $F(w_{Z_n})$ are assumed to average out sufficiently within the expectation required to calculate $\mathcal{C}_{P|Z_n}$. This allows the approximation $\mathcal{C}_{P|Z_n} \approx s^2\mathrm{Tr}(F(w_{Z_n}))$.

**Approximation of $\mathcal{C}_P$** We first consider the conditional inconsistency for a fixed $Z_n$, denoted $\mathcal{C}_{P|Z_n}$, by removing the outer expectation $\mathbb{E}_{Z_n}$:

$$\mathcal{C}_{P|Z_n} = \mathbb{E}_{\Theta,\Theta' \sim \Theta_{P|Z_n}} \mathbb{E}_X[\mathrm{KL}(f(\Theta, X)\|f(\Theta', X))]$$

Applying the isotropic covariance posterior assumption, $\Theta = w_{Z_n} + \delta$ and $\Theta' = w_{Z_n} + \delta'$, where $\delta, \delta'$ are independent perturbations ($\mathbb{E}[\delta] = \mathbb{E}[\delta'] = 0, \mathrm{Cov}[\delta] = \mathrm{Cov}[\delta'] = s^2 \mathbf{I}_d$).

$$\mathcal{C}_{P|Z_n} \approx \mathbb{E}_{\delta,\delta'} \mathbb{E}_X[\mathrm{KL}(f(w_{Z_n} + \delta, X)\|f(w_{Z_n} + \delta', X))]$$

Using the second-order Taylor expansion for KL divergence taking the expectation over $X$, valid for small $\|\delta - \delta'\|$ (i.e., small $s^2$):

$$\mathbb{E}_X[\mathrm{KL}(f(w_{Z_n} + \delta, X)\|f(w_{Z_n} + \delta', X))] = \frac{1}{2}(\delta - \delta')^T F(w_{Z_n} + \delta')(\delta - \delta') + O(\|\delta\|^3)$$

Let $u = \Theta - \Theta' = \delta - \delta'$. Since $\delta, \delta'$ are independent, $u \sim \mathcal{N}(0, 2s^2 \mathbf{I}_d)$. Substituting this into the expression for $\mathcal{C}_{P|Z_n}$:

$$
\begin{aligned}
\mathcal{C}_{P|Z_n} &= \mathbb{E}_u\left[\frac{1}{2}u^T F(\Theta')u\right] + O(\|\delta\|^3) \\
&= \mathbb{E}_u\left[\frac{1}{2}u^T F(w_{Z_n})u\right] + O(\|\delta\|^3) \quad \text{(FIM Constancy in Expectation Assumption)} \\
&= \frac{1}{2}\mathrm{Tr}(\mathrm{Cov}(u)F(w_{Z_n})) + \frac{1}{2}\mathbb{E}[u]^T F(w_{Z_n})\mathbb{E}[u] + O(\|\delta\|^3) \\
&= \frac{1}{2}\mathrm{Tr}(2s^2 \mathbf{I}_d F(w_{Z_n})) + 0 + O(\|\delta\|^3) \quad (\mathbb{E}[u] = 0) \\
&\approx s^2 \mathrm{Tr}(F(w_{Z_n}))
\end{aligned}
$$

Thus, the conditional inconsistency for a fixed $Z_n$ is approximately proportional to the trace of the FIM evaluated at $w_{Z_n}$:

$$\mathcal{C}_{P|Z_n} \approx s^2 \mathrm{Tr}(F(w_{Z_n})) \tag{12}$$

The overall inconsistency $\mathcal{C}_P$ is the expectation of this quantity over $Z_n$: $\mathcal{C}_P \approx \mathbb{E}_{Z_n}[s^2 \mathrm{Tr}(F(w_{Z_n}))]$.

**Approximation of $S_\rho(w_{Z_n})$** Applying the same second-order KL approximation to the definition of $S_\rho(w_{Z_n})$:

$$S_\rho(w_{Z_n}) = \max_{\|\delta\|_2 \le \rho} \frac{1}{2}\delta^\top F(w_{Z_n})\delta + O(\|\delta\|^3)$$

The maximum value of the quadratic form $\delta^T A \delta$ for a positive semi-definite matrix $A$ subject to $\|\delta\|_2 \le \rho$ is achieved when $\delta$ is aligned with the eigenvector corresponding to the largest eigenvalue ($\lambda_{\max}(A)$) and has norm $\rho$. Thus:

$$S_\rho(w_{Z_n}) = \frac{1}{2}\rho^2 \lambda_{\max}(F(w_{Z_n})) \tag{13}$$

This shows that the local sensitivity $S_\rho$ is approximately proportional to the largest eigenvalue of the FIM.

**Connecting $\mathcal{C}_{P|Z_n}$ and $S_\rho(w_{Z_n})$** For a $d \times d$ positive semi-definite matrix $A$, the relationship between its trace and largest eigenvalue is given by $\frac{1}{d}\mathrm{Tr}(A) \le \lambda_{\max}(A) \le \mathrm{Tr}(A)$. Applying this to the FIM $F(w_{Z_n})$:

$$\frac{1}{d}\mathrm{Tr}(F(w_{Z_n})) \le \lambda_{\max}(F(w_{Z_n})) \le \mathrm{Tr}(F(w_{Z_n}))$$

Substituting this into the approximation for $S_\rho(w_{Z_n})$ from Eq. equation 13:

$$\frac{\rho^2}{2d}\mathrm{Tr}(F(w_{Z_n})) \le S_\rho(w_{Z_n}) \le \frac{\rho^2}{2}\mathrm{Tr}(F(w_{Z_n}))$$

Let's assume a plausible connection, for instance, $s^2 = \rho^2/d$. Substituting this into the approximation for $\mathcal{C}_{P|Z_n}$ from Eq. (12), we get $\mathcal{C}_{P|Z_n} \approx \frac{\rho^2}{d}\text{Tr}(F(w_{Z_n}))$. Combining this with the bounds for $S_\rho(w_{Z_n})$:

$$\frac{1}{2}\left(\frac{\rho^2}{d}\text{Tr}(F(w_{Z_n}))\right) \leq S_\rho(w_{Z_n}) \leq \frac{d}{2}\left(\frac{\rho^2}{d}\text{Tr}(F(w_{Z_n}))\right)$$

This leads to the final approximate relationship between the conditional inconsistency (for a fixed $Z_n$) and the local sensitivity (at the corresponding $w_{Z_n}$):

$$\frac{1}{2}\mathcal{C}_{P|Z_n} \leq S_\rho(w_{Z_n}) \leq \frac{d}{2}\mathcal{C}_{P|Z_n} \tag{14}$$

This result suggests that, under the stated assumptions, the conditional inconsistency $\mathcal{C}_{P|Z_n}$ and the local sensitivity $S_\rho(w_{Z_n})$ are approximately proportional, with the proportionality factor potentially depending on the parameter dimension $d$.

**anisotropic covariance** Let $\text{Cov}[\Theta_{P|Z_n}] = s^2\Sigma$, where $s^2 = \frac{\rho^2}{d}$. Starting from $\mathcal{C}_{P|Z_n} = \frac{1}{2}\text{Tr}(\boldsymbol{\Sigma}F(w_{Z_n}))$,

$$\lambda_{min}(\Sigma)\text{Tr}(F) \leq \text{Tr}(\Sigma F) \leq \lambda_{\max}(\Sigma)\text{Tr}(F)$$

$$\lambda_{min}(\Sigma)\lambda_{\max}(F) \leq \text{Tr}(\Sigma F) \leq \lambda_{\max}(\Sigma)d\lambda_{\max}(F)$$

$$\frac{\rho^2}{2d\lambda_{\max}(\Sigma)}\text{Tr}(\Sigma F) \leq \frac{\rho^2}{2}\lambda_{\max}(F) \leq \frac{\rho^2}{2\lambda_{min}(\Sigma)}\text{Tr}(\Sigma F)$$

$$\frac{1}{\lambda_{\max}(\Sigma)}\mathcal{C}_{P|Z_n} \leq S_\rho(w_{Z_n}) \leq \frac{d}{\lambda_{min}(\Sigma)}\mathcal{C}_{P|Z_n}$$

**Practical Considerations: Eigenvalue Spectrum of Neural Networks**  In practice, for deep learning models, the FIM often exhibits a sparse eigenvalue spectrum: many eigenvalues are close to zero, and only a few are significantly large. In such cases:

- The trace $\text{Tr}(F) = \sum \lambda_i$ is dominated by the sum of the few large eigenvalues.
- The ratio $\lambda_{\max}(F)/\text{Tr}(F)$ might be closer to $1/m'$ than $1/d$, where $m' \ll d$ is the "effective rank" or number of dominant eigenvalues.

This implies that the bounds relating $\lambda_{\max}(F)$ and $\text{Tr}(F)$ might be tighter than the general $1/d$ and $1$ factors suggest. Consequently, the relationship between $\mathcal{C}_{P|Z_n}$ (related to trace) and $S_\rho$ (related to max eigenvalue) could be closer to direct proportionality than Eq. equation 5 indicates, especially if $s^2$ is appropriately related to $\rho^2$.

**Summary and Limitations**  This analysis provides a heuristic argument suggesting a connection between conditional inconsistency $\mathcal{C}_{P|Z_n}$ and local sensitivity $S_\rho(w_{Z_n})$. Under assumptions of a Gaussian posterior, small variance $s^2$, validity of second-order KL approximations, local FIM constancy, and a specific link between $s^2$ and $\rho^2$ (e.g., $s^2 = \rho^2/d$), we find that $S_\rho(w_{Z_n})$ is approximately proportional to $\mathcal{C}_{P|Z_n}$, potentially up to a factor related to dimension $d$. This connection is mediated by the trace and the maximum eigenvalue of the Fisher Information Matrix. The practical observation of sparse FIM eigenvalues might strengthen this relationship.

## C  DECISION BOUNDARY OF NEURAL NETWORKS AND PRINCIPAL EIGENSPACE OF FIM

To intuitively analysis the role of $\delta_1$ in training of neural network, we conducted experiments using 3-layer fully-connected neural network on two-dimensional synthetic data. the data is generated from a mixture of three Gaussian distributions, a setup analogous to that employed by Jang et al. (2022) in their investigation of the characteristic of the FIM eigensubspace. Their work demonstrated that perturbing parameters along the principal eigenvectors of the FIM can lead to significant modifications in the decision boundary, such as increasing or decreasing the margins of specific classes.

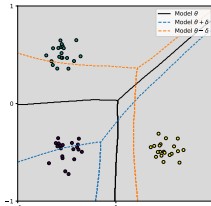 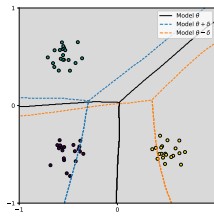 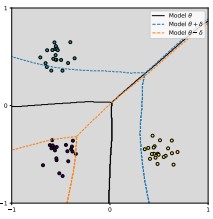 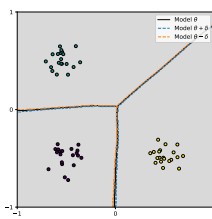

(a) Decision boundary perturbed by $\delta_1$ from $\varepsilon_1$

(b) Decision boundary perturbed by $\delta_1$ from $\varepsilon_2$

(c) Decision boundary perturbed by $\delta_1$ from $\varepsilon_3$

(d) Decision boundary perturbed by $\varepsilon$

Figure 4: A synthetic classification example. the black, blue, orange lines correspond to decision boundaries of the NN with trained parameter values, and parameter values perturbed by $\delta_1$. Each plot use different noise.

Our investigation focuses on whether $\delta_1$, despite being derived from only a single gradient step (as described in Algorithm 1) and thus influenced by an initial random noise vector $\varepsilon$, still induces substantial changes in the neural network's decision boundary. Figure 4 visualizes these effects. The black lines in each subfigures depict the original decision boundary obtained with the trained parameters $w$. Figure 4 (a-c) show the perturbed decision boundaries (blue and orange lines) when distinct $\pm\delta_1$ with $\rho = 0.5$ is added to $w$. Each of these $\delta_1$ vectors was computed using a different random initialization noise vector, denoted as $\varepsilon_1$, $\varepsilon_2$, and $\varepsilon_3$, respectively. For a direct comparison of the perturbation's nature, Figure 4(d) illustrates the decision boundary perturbed by directly adding the random noise vector $\varepsilon$ to $w$. This vector $\varepsilon$ is sampled from same distribution as initial vectors (e.g.$\varepsilon_1$) and, is scaled to $\|\varepsilon\|_2 = \rho$ same with $\delta_1$. As observed in Figure 4 (d), direct perturbation with such an arbitrary random noise vector does not meaning fully alter the decision boundary, even when its norm is equivalent to that of the $\delta_1$. This is sharply opposed with the significant changes induced by $\delta_1$ perturbations shown in Figures 4 (a-c), underscoring that the direction derived by Algorithm 1, even in a single step, is substantially more influential than arbitrary noise of the same magnitude. This result intuitively suggest that the perturbation $\delta_1$ with single gradient step still meaningful and aligning with principle eigen vectors of FIM.

To investigate the alignment between the single-step perturbation vector $\delta_1$ and principle eigenspace of FIM, we explicitly calculate the FIM and its top three eigenvector $v_1$, $v_2$, and $v_3$, corresponding to largesst eigenvalues $\lambda_1 > \lambda_2 > \lambda_3$. The perturbation $\delta_1$, results from one normalized gradient ascent step applied to the KL divergence objective, starting from an initial random noise $\varepsilon$. In terms of power iteration algorithm, the $\delta_1$ after first iteration without normalization, is sum of eigenvector of FIM weighted by $\lambda_i \alpha_i$.

Formally, let the initial random noise $\varepsilon$ be expressed in the eigenbasis of $F(w)$ as $\varepsilon = \sum_i^m \alpha_i v_i$. $\varepsilon \sim \mathcal{N}(0, \sigma^2 I_m)$, then the coefficient $\alpha_i$ are i.i.d. as $\mathcal{N}(0, \sigma^2)$ since $\{v_i\}$ form an orthonormal basis.

$$F(\theta)\varepsilon = \sum_i^m \lambda_i v_i v_i^\top \sum_i^m \alpha_i v_i$$

$$= \sum_i^m \lambda_i \alpha_i v_i$$

So cosine similarity between $\delta_1$ and $v_i$ is $\lambda_i \alpha_i$. And $\frac{\|\sum_i^3 \delta_1^T v_i\|}{\|\delta_1\|}$, which indicates how much the $\delta_1$ is in principle eigen space, $\{u | u = av_1 + bv_2 + cv_3, \quad abc \in [0,1]\}$ of FIM, is $\frac{\|\sum_i^3 \alpha_i \lambda_i v_i\|}{\|\delta\|}$.

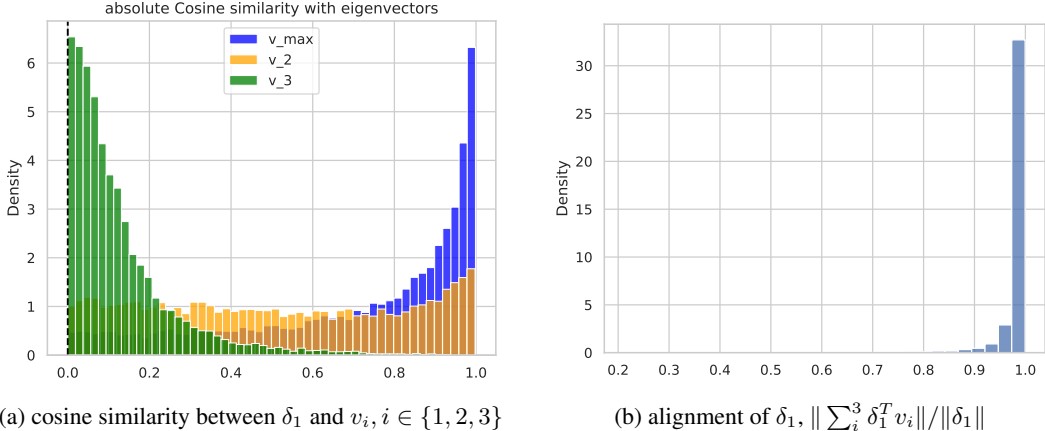

(a) cosine similarity between $\delta_1$ and $v_i, i \in \{1, 2, 3\}$

(b) alignment of $\delta_1$, $\|\sum_i^3 \delta_1^T v_i\|/\|\delta_1\|$

Figure 5: A synthetic classification example. $\delta_1$ are align with top three eigen Vector of FIM sampling from 10000 gaussian noises $\varepsilon$

Figure 5 presents empirical results from this analysis. Figure 5 (a) shows histograms of the absolute cosine similarities between $\delta_1$ (generated from 10,000 different $\varepsilon$ samples) and each of the top three eigenvectors $v_1$, $v_2$, and $v_3$. We observe that $\delta_1$ tends to have a higher cosine similarity with $v_1$ (corresponding to the largest eigenvalue $\lambda_1$) compared to $v_2$, and $v_3$. Furthermore, Figure 5 (b) displays the distribution of the squared norm of the projection of $\delta_1$ onto the top-3 eigenspace. The values are predominantly close to 1, indicating that $\delta$ vectors derived from different initial noise samples are largely confined to this principal subspace. These results empirically support the theoretical expectation that the single-step perturbation $\delta_1$ is predominantly aligned with the principal eigenspace of the FIM.

# D EXTRA EXPERIMENTS

## D.1 ANALYSIS OF LOCAL INCONSISTENCY ESTIMATION

In this section, we analyze how the estimation of local inconsistency $S_\rho(\theta)$ is affected by approximation choices: the mini-batch size used for perturbation ($m$-sharpness) and the number of gradient ascent steps ($K$). Regarding $m$-sharpness, we follow the protocol introduced for SAM—computing *independent* perturbations on disjoint sub-batches in parallel and *averaging* the perturbed gradients for the update—and replicate this scheme for IAM-S. $m$ indicate the size of disjoint sub-batch.

### D.1.1 EFFECT OF INNER MAXIMIZATION STEPS ($K$)

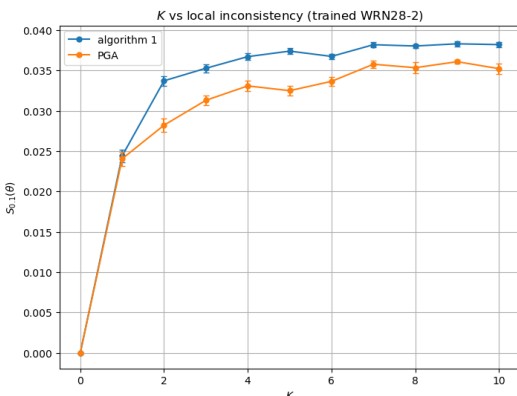

Figure 6: Estimated $S_\rho(\theta)$ with respect to $K$ on WRN28-2 (CIFAR-10) using Algorithm 1 vs. Projected Gradient Ascent. Algorithm 1 with $K = 3$ offers a sufficient approximation of the true maximizer.

We investigate the impact of the number of steps $K$ used in Algorithm 1 on model performance. From the perspective of estimating $S_\rho(\theta)$, increasing $K$ naturally yields a more accurate approximation of the worst-case perturbation $\delta^*$ and, consequently, a tighter lower bound on the local inconsistency, as illustrated in Figure 6. This suggests that a more precise estimation of $S_\rho(\theta)$ (i.e., using $K > 1$) during training may lead to better regularization and improved generalization.

To verify this, we conducted an ablation study on $K$ using IAM-D trained on CIFAR-10/100, following the standard hyperparameters described in Appendix E. The results are summarized in Table 4.

As shown in Table 4, we observe a consistent improvement in generalization performance as $K$ increases. Specifically, increasing $K$ from 1 to 3 reduces the test error from 3.28% to 2.99%, although this comes at the cost of increased computational overhead.

Notably, this finding stands in contrast to SAM, where increasing the number of inner maximization steps was reported to have no strong effect on test accuracy for CIFAR-10. While SAM found that a single step was sufficient to obtain a good approximation of the maximizer, our results indicate that for IAM, a more accurate estimation of the local inconsistency via multiple steps ($K > 1$) provides tangible benefits to the final model performance.

Table 4: Test error and training cost of IAM with respect to $K$ and $m$ (WRN28-10).

| $K$ | CIFAR-10 | | | CIFAR-100 | | | Running time |
|---|---|---|---|---|---|---|---|
| | Standard | $m = 32$ | $m = 16$ | Standard | $m = 32$ | $m = 16$ | (s/epoch) |
| 1 | $3.28_{\pm 0.06}$ | $3.05_{\pm 0.02}$ | $3.03_{\pm 0.02}$ | $17.16_{\pm 0.03}$ | $16.92_{\pm 0.04}$ | $16.58_{\pm 0.05}$ | 239 (1.0×) |
| 2 | $3.03_{\pm 0.02}$ | $2.85_{\pm 0.04}$ | $\mathbf{2.80}_{\pm 0.02}$ | $16.92_{\pm 0.04}$ | $16.08_{\pm 0.08}$ | $15.45_{\pm 0.09}$ | 311 (1.3×) |
| 3 | $\mathbf{2.99}_{\pm 0.04}$ | $2.86_{\pm 0.01}$ | $2.91_{\pm 0.02}$ | $16.90_{\pm 0.03}$ | $15.89_{\pm 0.01}$ | $15.34_{\pm 0.05}$ | 378 (1.6×) |
| 5 | $\mathbf{2.98}_{\pm 0.03}$ | $\mathbf{2.80}_{\pm 0.08}$ | $2.87_{\pm 0.01}$ | $\mathbf{16.62}_{\pm 0.02}$ | $\mathbf{15.78}_{\pm 0.18}$ | $\mathbf{15.26}_{\pm 0.01}$ | 525 (2.2×) |

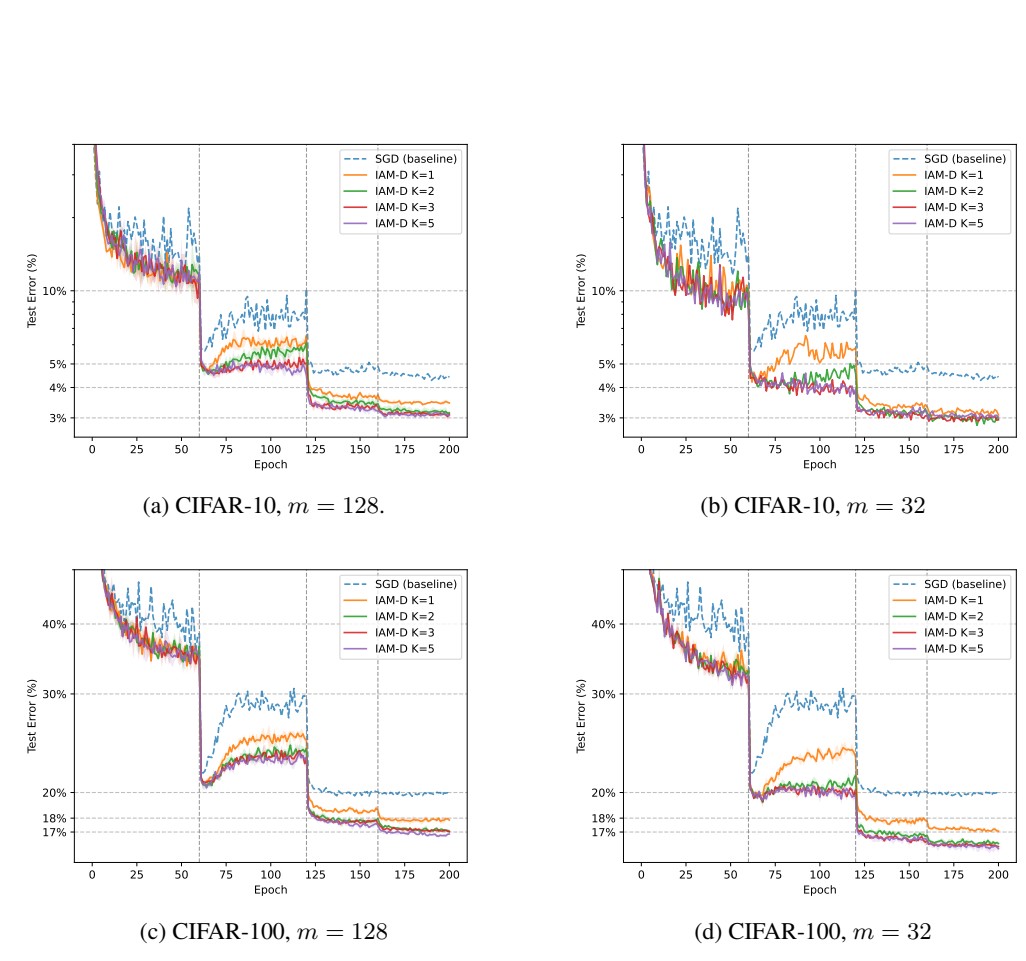

(a) CIFAR-10, $m = 128$.

(b) CIFAR-10, $m = 32$

(c) CIFAR-100, $m = 128$

(d) CIFAR-100, $m = 32$

Figure 7: The evolution of test error (log-scale) with SGD and IAM-D according to different sub-batch size and K

### D.1.2   M-SHARPNESS IN IAM: PARALLEL PER-SUB-BATCH PERTURBATIONS

On CIFAR-10 with a fixed total batch size 256, we split each batch into sub-batch size $m \in \{4, 16, 64, 256\}$, compute perturbation $\delta$ and gradient $\nabla_\theta L(\theta + \delta)$ on each mini-batch and update with the mean of gradients. We sweep $\rho \in \{0.005, 0.01, 0.02, 0.05, 0.1, 0.2, 0.5\}$ and repeat each $(m, \rho)$ condition three times with independent seeds. All other training details (backbone, schedule, preprocessing) are identical to the main IAM-S experiments in the paper.

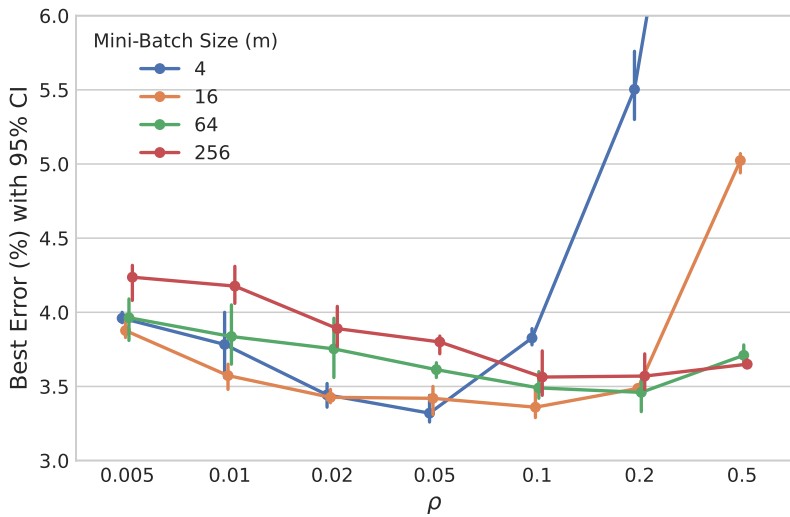

Figure 8: Test error as a function of $\rho$ for different values of $m$.

Figure 8 shows that smaller values of $m$ tend to yield models having better generalization ability as observed in Foret et al. (2021).

## D.2 SUPERVISED LEARNING

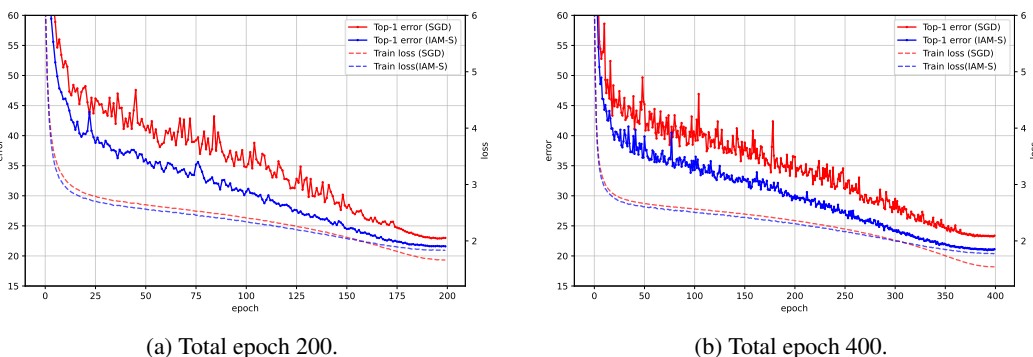

(a) Total epoch 200.  (b) Total epoch 400.

Figure 9: The evolution of test error with SGD and IAM-S when training ResNet on Imagenet.

Table 5: Top-{1, 5} error (mean ± stderr) of SGD, SAM, and IAM-S trained with ImageNet.

| Epoch | SGD | | SAM | | IAM-S | |
|---|---|---|---|---|---|---|
| | Top-1 | Top-5 | Top-1 | Top-5 | Top-1 | Top-5 |
| 100 | $23.27_{\pm 0.08}$ | $6.72_{\pm 0.03}$ | $\mathbf{22.69}_{\pm 0.13}$ | $\mathbf{6.41}_{\pm 0.03}$ | $22.84_{\pm 0.04}$ | $6.46_{\pm 0.06}$ |
| 200 | $22.66_{\pm 0.12}$ | $6.51_{\pm 0.07}$ | $21.80_{\pm 0.12}$ | $5.99_{\pm 0.04}$ | $\mathbf{21.72}_{\pm 0.07}$ | $\mathbf{5.89}_{\pm 0.02}$ |
| 400 | $22.80_{\pm 0.23}$ | $6.66_{\pm 0.06}$ | - | - | - | - |

## D.3 FINE-TUNING VIT

To demonstrate the versatility of IAM beyond CNN architectures, we conducted additional fine-tuning experiments on ViT-S/16 pre-trained on ImageNet-1K using the CIFAR-10 dataset. We compared IAM-D against SGD and SAM.

We fine-tuned the models for 10,000 steps with a batch size of 128, with base optimizer SGD. Gradient clipping with max norm $= 1.0$ is applied. The initial learning rate was set to 0.01 with a linear decay schedule after 500 warmup steps. For perturbation magnitude $\rho$, we used $\rho = 0.05$ for SAM and $\rho = 0.1$ for IAM-D.

Table 6: Test error $\pm$ stderr of SGD, SAM, and IAM-D when fine-tuning ViT-S/16 on CIFAR-10.

| Method | Test error |
|---|---|
| SGD | $1.86_{\pm 0.01}$ |
| SAM | $1.56_{\pm 0.01}$ |
| IAM-D | $1.52_{\pm 0.02}$ |

IAM-D is consistently competitive to SAM on the transformer-based architecture, confirming that our proposed local inconsistency measure is effective across different model inductive biases.

## D.4 SEMI-SUPERVISED LEARNING

If we restrict SAM only to the **labeled** loss to avoid instability, we only minimize sharpness for the very small subset of labeled data (e.g., 250 samples). This fails to regularize the global landscape. To confirm this, we ran "FixMatch + SAM" on CIFAR-10 (250 labels).

This is significantly worse than FixMatch + SGD (6.26 %) and FixMatch + IAM-D (5.30 %). This failure case underscores the strength of IAM-D: it calculates inconsistency on unlabeled data without relying on potentially incorrect pseudo-labels, making it naturally superior for SSL.

Table 7: Test Error (mean ± stderr) on CIFAR-10 with 250 labels using a WRN-28-2 model.

| Method | Test error |
|---|---|
| FixMatch | $6.26_{\pm 0.39}$ |
| FixMatch + SAM | $9.90_{\pm 0.74}$ |
| FixMatch + IAM-D | $5.30_{\pm 0.08}$ |

# E  EXPERIMENTAL DETAILS

**Practical Considerations in estimating $S_\rho(\theta)$**

- **Computational Efficiency:** Calculating the FIM explicitly and performing eigenvalue decomposition is computationally expensive ($O(m^2)$ or worse, where $m$ is the number of parameters). Algorithm 1 avoids this by requiring only $K$ gradient computations (forward and backward passes) per estimation, making its computational cost approximately $O(mK)$, which is significantly more feasible for large networks.

- **Number of Steps (K):** Empirical studies on neural network Hessians and FIMs suggest that the eigenspectrum is often dominated by a huge largest eigenvalues. Thus, the Power Iteration method can converge quickly to the dominant eigenvector. In practice, using a small number of steps, often just $K = 3$, is found to be sufficient to get a reasonable estimate of the maximizing direction. This makes the computation highly efficient.

- **Averaging for reduce Variance from initialization:** The estimate of $S_\rho(w)$ obtained from Algorithm 1 depends on the random initialization $\delta_0$ with just $K = 1$. To obtain a more stable estimate, we compute the metric multiple times (e.g., 10 times) with different random initializations for $\delta_0$ and report the average value: $\mathbb{E}_{\delta_0}[\text{Estimate from Alg 1}]$.

**Infrastructure** Experiments are implemented in PyTorch 2.5.1 and executed on NVIDIA A40, A100 and L4 GPUs.

## E.1  EXPERIMENTAL DETAILS FOR FIGURE 1 (SECTION 4.6)

We trained 6CNN and WRN28-2 using SGD to investigate the relationship between generalization gap and local inconsistency. For 6CNN, each hyperparameter combination was run with 5 independent random seeds to assess variability. $\text{Tr}(H)$ and $\lambda_{\max}(H)$ were computed on a subset of 2,000 training examples, and $S_\rho$ was computed on a 5,000-sample unlabeled held-out set.

Table 8: Hyperparameters used for 6CNN and WRN28-2 on CIFAR-10.

| Hyperparameter | 6CNN | WRN28-2 |
|---|---|---|
| Dataset | CIFAR-10 | CIFAR-10 |
| Training data size | 45K | 45K |
| Initial learning rate | {0.001, 0.002, 0.005, 0.01, 0.02, 0.05} | {0.1, 0.03, 0.01} |
| Batch size | {32, 64, 128, 256, 512} | {32, 64, 128, 256, 512} |
| Weight decay | {0.0, $10^{-4}$, $5 \times 10^{-4}$, $10^{-3}$} | {0.0, $10^{-4}$, $5 \times 10^{-4}$} |
| Learning rate scheduling | constant | {cosine annealing, multi-step} |
| Data augmentation | False | {True, False} |
| Label smoothing | – | – |
| Epochs | until convergence ($< 400$) | {150, 200, 300} |
| $K$ | 3 | 1 |

## E.2  IMAGE CLASSIFICATION

Each reported metric is the mean± standard error computed over minimum test error from three independent runs.

**Dataset.** We evaluate on **CIFAR-10** (50,000 training, 10,000 test images), **CIFAR-100** (50,000 training, 10,000 test images), Fashion-MNIST, and SVHN (no additional datasets). CIFAR-10, CIFAR-100, and SVHN are resized to $32 \times 32$ and preprocessed with RandomCrop(32, padding= 4). Fashion-MNIST is preprocessed with RandomCrop(28, padding= 4). Below are applied augmentations in common:

- *RandomHorizontalFlip*($p = 0.5$), and

- *Normalization* using the official mean and standard deviation.

No additional augmentation such as Cutout or Mixup is applied.

**Optimization.** The models are trained for **200 epochs** with mini-batch size **128**. We use SGD with momentum 0.9, weight decay $5 \times 10^{-4}$ as an optimizer, and a multistep learning rate schedule that decays the initial rate 0.1 (0.01 for SVHN) by 0.2 at epochs 60, 120, and 160. We report the best score achieved by each SGD training run across either the standard epochs or the doubled epochs.

**Hyperparameters.** For image classification task, $\beta, \rho$ are tuned via grid search over $\beta \in \{0.1, 1.0, 5.0, 10.0, 20.0\}, \rho \in \{0.01, 0.05, 0.1, 0.5, 1.0\}$ with validation split using 10% of the training dataset. As seen in Figure, the best pairs are $(1.0, 0.1)$ for CIFAR-10 and $(10.0, 0.1)$ for CIFAR-100. For both datasets, $\beta$ and $\rho$ had a trade-off relation.

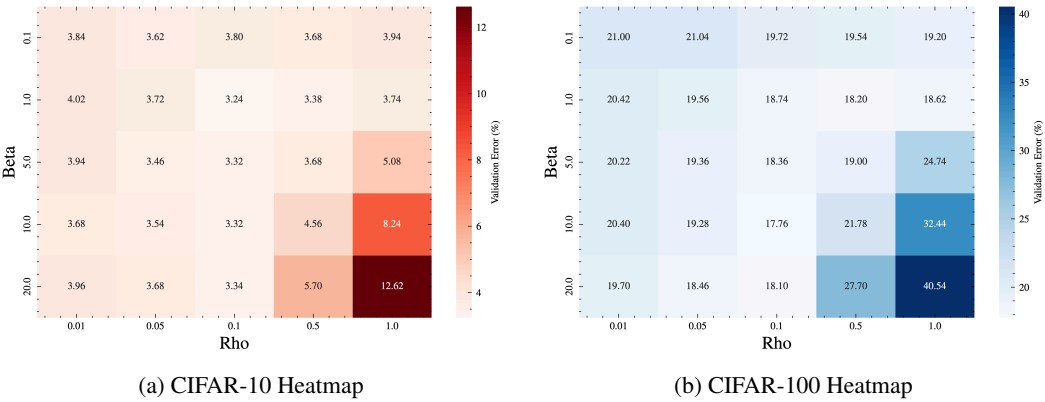

(a) CIFAR-10 Heatmap  (b) CIFAR-100 Heatmap

**Loss function.** Cross-entropy with label smoothing ($\alpha = 0.1$) is used for all methods.

### E.3 Semi-supervised Learning

In semi-supervised learning experiment, we shared most of the settings with image classification. Each reported metric is computed over minimum test error from three independent runs. Experiments with FixMatch are stated in a separate section.

**Optimization.** Models are trained for **200 epochs** without learning rate scheduling.

**Hyperparameters.** We used $\beta = 1.0$ and $\rho = 0.1$ for CIFAR-10 and $\beta = 10.0$ and $\rho = 0.1$ for CIFAR-100. SAM is also trained with $\rho = 0.1$. The batch size 128 is used for labeled data and 384 for unlabeled data.

**FixMatch.** We followed the reported FixMatch settings. WRN-28-2 for CIFAR-10, WRN-28-8 for CIFAR-100 are trained for $2^{20}$ **iterations** with SGD as the base optimizer using the learning rate 0.03, momentum 0.9, weight decay $5e - 4$, with cosine learning rate scheduling. For IAM-D, $\rho = 0.01, \beta = 1.0$ is applied for CIFAR-10 and $\rho = 0.05, \beta = 1.0$ is applied for CIFAR-100. The batch size for the labeled data was 64, and for unlabeled data was 448. We applied EMA with decay 0.99.

### E.4 Self-supervised learning

Each reported metric is the mean **test accuracy** obtained from three independent runs.

**Dataset.** We use the **CIFAR-10** benchmark. All images are resized to $32\times32$ and augmented with the SimCLR(Chen et al., 2020) pipeline:

- *RandomResizedCrop*$(32, \text{ scale}=(0.4, 1.0))$,
- *RandomHorizontalFlip*$(p = 0.5)$,
- *ColorJitter*$(0.4, 0.4, 0.2, 0.1)$ with probability $0.8$,
- *RandomGrayscale*$(p{=}0.2)$, and
- *Normalization* using the official mean and standard deviation.

**Encoder&Projection Head.** We adopt a **ResNet-18** backbone with the first convolution modified to $3\times3$ layer with stride $= 1$ and the max-pool removed. The projector is a two-layer MLP (hidden size $512$, output size$128$) with ReLU activation.

**Optimization.** Models are trained for **200 epochs** with mini-batch size **1024**. We use SGD (momentum 0.9, weight decay $1\times10^{-4}$) and a cosine-annealing learning-rate schedule starting at $1.0$ after a 10-epoch warm-up.

**Contrastive Loss.** The NT-Xent loss is computed with temperature $\tau{=}0.5$.

**IAM Hyperparameters.** We set the inconsistency weight $\beta{=}1.0$, neighborhood radius $\rho{=}0.1$, and noise-scale $3.0$ (Gaussian initialization). The local inconsistency is computed between projection head outputs with temperature $\tau{=}0.5$.

**Stability Heuristics.** It is identical to image classification setting.

**Linear Evaluation.** After every 5 epochs (and at the final epoch), a frozen encoder is evaluated via a linear probe trained for 20 epochs with AdamW optimizer on the full training set (batch size 1024). The reported metric is the probe's test accuracy.

## F  LLM usage

We use LLMs solely for language polishing (grammar, phrasing, and minor style edits). No private or unpublished data were provided to the tool. All scientific content and claims are our own, and the authors take full responsibility for the manuscript.

