# OpenReview forum: "Inconsistency-Aware Minimization: Improving Generalization with Unlabeled Data"
_ICLR.cc/2026/Conference — Submitted to ICLR 2026_

### Official Review · Reviewer_aam4 · 2025-10-26

**Soundness:** 2
**Presentation:** 3
**Contribution:** 2
**Rating:** 4
**Confidence:** 3

**Summary:**

The paper defines local inconsistency as the change in a model’s output distribution under small parameter perturbations, shows it can be computed with a single trained model and unlabeled data, and links it to the Fisher Information Matrix and loss curvature. It introduces Inconsistency-Aware Minimization (IAM) in two variants (IAM-D, IAM-S) and provides a generalization bound that replaces a Hessian term with the top FIM eigenvalue under near-interpolation (Theorem 4.1). Experiments show that the inconsistency measure correlates with generalization (Figure 1), IAM improves over SGD/SAM/ASAM on vision benchmarks (Table 1), gives ImageNet gains with ResNet-50 (Table 2), and benefits semi/self-supervised setups including FixMatch and SimCLR (Table 3, Figure 3).

**Strengths:**

1. The local inconsistency can be computed from a single trained model using only unlabeled data. It is directly differentiable, enabling optimization within standard training pipelines.
2. The paper establish a generalization bound (Theorem 4.1), validating the single-model approximation theoretically
3. Algorithm 1 estimates the local inconsistency using power iteration updates without explicit eigen decomposition, achieving linear complexity versus quadratic. This makes the method computationally practical for large networks.
4. The method shows consistent improvements across supervised learning (Tables 1,2), semi-supervised with limited labels (Table 3), and self-supervised SimCLR (Figure 3). Success across multiple paradigms suggests genuine generality.

**Weaknesses:**

1. Theorem 4.1 assumes near-interpolation and many other assumptions. The bound validity remains unclear if these assumptions are violated. The paper didn't illustrate why the assumptions are reasonable, and also didn't show whether the assumptions are valid in practical training.
2. Algorithm 1 use K=1 for efficiency, but the paper provides no empirical verification on why K=1 is enough.
3. In Table 2, ImageNet results compare only against SGD baseline with no SAM or ASAM comparison, despite SAM being the primary supervised-learning baseline on CIFAR datasets (Table 1)

**Questions:**

1. Table 2 shows IAM-S results for ResNet-50 at 100 and 200 epochs but lists "-" for 400 epochs with no explanation—did training fail, diverge, or was it simply not run?
2. What about the computational efficiency compared to baselines?
3. All results are on CNN type model. Could you test the transformers like ViT?

---

> ### Author Response · Authors · 2025-11-21
>
> > Theorem 4.1 assumes near-interpolation and many other assumptions. The bound validity remains unclear if these assumptions are violated. The paper didn't illustrate why the assumptions are reasonable, and also didn't show whether the assumptions are valid in practical training.
>
> We would like to clarify that the near-interpolation assumption is not merely a convenient hypothesis but a condition that is both theoretically justified by the structure of the loss function and empirically prevalent in modern deep learning.
>
> #### 1. Theoretical Justification (Vanishing Residual Term):
>
> As detailed in **Lemma A.1** of Appendix A, the Hessian of the empirical loss $H_S(θ)$ decomposes into the Fisher Information Matrix $F_S(θ)$ and a residual term $R_S(θ)$:
>     $$H_S(θ) = F_S(θ) + R_S(θ)$$
> Crucially, the residual term $R_S(θ)$ is explicitly dependent on the prediction error.
> $$R_S(θ) = \sum_{i=1}^N \sum_{k=1}^C (f(x_i;θ)-y_i)\_{k} \nabla_θ^2 z_k(x_i;θ)$$
> As shown in the proof of Lemma A.1, the per-sample residual is proportional to the difference between the model prediction and the label: $(f(x_i; θ) - y_i)$.
> Therefore, in the near-interpolation regime where **the training loss approaches zero** (i.e., $f(x_i; θ) \approx y_i$), the residual term $R_S(θ)$ mathematically vanishes (i.e., $\epsilon_R \to 0$). This ensures that $λ_{max}(H_S(θ)) \approx λ_{max}(F_S(θ))$, validating the substitution used in our bound.
>
> #### 2. Validity in practical deep learning:
>
> The near-interpolation assumption is consistent with the standard behavior of over-parameterized deep neural networks. As demonstrated in the seminal work by Zhang et al. (2017) (*Understanding deep learning requires rethinking generalization*), modern deep networks possess sufficient effective capacity to memorize the training set completely, even when labels are randomized.
> In our experiments, we utilize standard over-parameterized architectures (e.g., ResNet, WideResNet), which typically achieve near-zero training error (interpolation) upon convergence. Consequently, the assumption that $\epsilon_R \approx 0$ holds in the standard training scenarios discussed in our work.
>
>
> #### 3. Standard assumptions inherited from prior art
> We respectfully point out that the additional assumptions are neither novel nor restrictive constraints introduced by our specific method. Rather, they are standard assumptions inherited directly from prior art, specifically Luo et al. (2024), upon which our proof is built. These conditions—such as the boundedness of higher-order derivatives—are standard in deriving Hessian-based generalization bounds and are necessary to guarantee the validity of the concentration inequalities used in the proof.
>
>
> ---
> > Algorithm 1 use K=1 for efficiency, but the paper provides no empirical verification on why K=1 is enough.
>
>
> - IAM with $K=1$ still shows competitive performance against strong baseline.
>
> First of all, we also use $K > 1$ for estimating $S_\rho(θ)$ as a metric for figure 1. As described in Appendix E, $K=3$ shows to be sufficient to get a reasonable estimate of the maximizing direction. See figure 6 in the revised paper Appendix 3.1.1 for convergence performance of Algorithm 1 vs. usual projected gradient ascent step.
>
> For optimization (IAM), Algorithm 1 with $K$ needs $K$ additional backward passes for each step.
> Since SAM (or its variant) requires one additional backward per each step, we employ $K=1$  for matching computation efficiency with SAM and ASAM.
> Foret et al. (2021) reported that the test accuracy of SAM is not strongly affected by the number of inner maximization iterations.
> In SAM, the test accuracy gain from increasing the number of inner maximization iterations was marginal (0.01-0.04 pp in CIFAR-10 and 0.1-0.13 pp in CIFAR-100).
> Thus, it is reasonable to report reults with one inner maximization step for IAM ($K=1$) and SAM.
>
> To verify whether the more precise approximation helps generalization, we conduct ablation study on IAM with $K>1$. The results stand in contrast to those for SAM reported in Foret et al. (2021), showing improved generalization. See Appendix D.1 for the detailed results and the empirical relation between $K$ and $m$-sharpness.

---

> ### Author Response · Authors · 2025-11-21
>
> ---
> > In Table 2, ImageNet results compare only against SGD baseline with no SAM or ASAM comparison, despite SAM being the primary supervised-learning baseline on CIFAR datasets (Table 1)
>
> First, we would like to clarify that the original purpose of our ImageNet experiment was primarily to demonstrate the **scalability** of our proposed method to large-scale datasets, as described in the last paragraph of Section 5.2.
>
> Our main goal was to verify that IAM remains computationally feasible and effective when applied to large-scale datasets and deeper architectures (such as ResNet-50), confirming that it consistently improves generalization over the standard SGD baseline.
>
> However, to address your concern, we have conducted an additional comparison against SAM. For this experiment, all algorithms (SGD, SAM, and IAM-S) were trained under the same settings, using a batch size of 1024 and a learning rate of 0.2.
>
> It is important to note that while the results for SAM were obtained using a $\rho$ value identified through hyperparameter tuning on ImageNet and ResNet-50 by Foret et al. (2021), the hyperparameters for IAM were **manually selected** due to limited computational resources for tuning. Consequently, this comparison places IAM at a disadvantage, as it compares a tuned baseline against a potentially suboptimal, untuned version of our method.
>
> Despite this unfavorable condition, IAM-S successfully outperformed both SGD and SAM, as shown in the table below:
>
> **Table**: Test error ± stderr of SGD, SAM, and IAM trained on ImageNet with 200 epochs.
> |     | Epoch | SGD|SAM|IAM-S|
> |-|-|-|-|-|
> |Top-1|200|22.66 ± 0.12|21.95 ± *|**21.72 ± 0.07**|
> |Top-5|200| 6.51 ± 0.06| 6.05  ± *|**5.90 ± 0.02**|
>
> Due to the high training costs associated with SAM on ImageNet, we currently report the result from a single run. However, we are in the process of completing two additional runs to ensure statistical rigor. We will include the averaged results over three independent trials in the revised version of the paper as soon as possible.
>
> ---
> > Q1. Table 2 shows IAM-S results for ResNet-50 at 100 and 200 epochs but lists "-" for 400 epochs with no explanation—did training fail, diverge, or was it simply not run?
>
> It was not run because the comparison was designed based on computational cost. As mentioned earlier, IAM requires an additional backward pass at each step, like SAM, compared to SGD, to compute the local inconsistency gradient. Therefore, to align the computational budget, we primarily focused on the 200-epoch setting for IAM, which incurs a computational cost comparable to a longer standard SGD training run.
>
> However, to address your inquiry, we have conducted a 400-epoch run for IAM-S. The results, including a single-run result for IAM-S at 400 epochs, are presented in the table below:
>
> Table: Top-{1, 5} error (mean ± stderr) of IAM-S and SGD trained on ImageNet.
> |Epoch|IAM-S Top-1|IAM-S Top-5|SGD Top-1|SGD Top-5|
> |-|-|-|-|-|
> |100|**22.84** ± 0.04|**6.46** ± 0.06|23.27 ± 0.08|6.72 ± 0.03|
> |200|**21.72** ± 0.07|**5.90** ± 0.02|22.66 ± 0.12|6.51 ± 0.07|
> |400|20.95|5.61|22.80 ± 0.23|6.66 ± 0.06|
>
> As shown in the table, IAM-S exhibits a distinct advantage over SGD in long-training regimes. Similar to the observations made by Foret et al. (2021) regarding SAM, IAM-S continues to improve performance as the total number of epochs increases, without succumbing to overfitting. In contrast, SGD shows signs of saturation or slight degradation at 400 epochs compared to 200 epochs, whereas IAM-S achieves a significant performance gain, reaching a top-1 error of 20.95%.
>
> ---
> > Q2. What about the computational efficiency compared to baselines?
>
> **Table**: Overhead comparison between optimization algorithms.
> |Optimizer|# of gradient computations|backward time (ms/step)|running time on ImageNet|
> |-|-|-|-|
> |SGD|1|49|943  s/epoch|
> |SAM|2|95|1613 s/epoch|
> |IAM|2|99|1878 s/epoch|
>
> ---
> > Q3. All results are on CNN type model. Could you test the transformers like ViT?
>
> To demonstrate the versatility of IAM beyond CNN architectures, we conducted additional fine-tuning experiments on ViT_s/16 pre-trained on ImageNet-1K using the CIFAR-10 dataset. We compared IAM-D against SGD and SAM.
>
> **Settings:**
> We fine-tuned the models for 10,000 steps with a batch size of 128, using SGD as the base optimizer. Gradient clipping with max_norm = 1.0 is applied. The initial learning rate was set to 0.01 with a linear decay schedule after 500 warm-up steps. For the perturbation magnitude $\rho$, we used $\rho=0.05$ for SAM and $\rho=0.1$ for IAM-D.
>
> **Table:** Test error ± stderr of SGD, SAM, and IAM-D when fine-tuning ViT-S/16 on CIFAR-10.
> |Method| |
> | --- | --- |
> |SGD|1.86 ± 0.01|
> |SAM|1.56 ± 0.01|
> |IAM-D|**1.52** ± 0.02|
>
> As shown above, IAM-D is consistently competitive with SAM on a transformer-based architecture, confirming that our proposed local inconsistency measure is effective across different model inductive biases.

---

### Official Review · Reviewer_HS3y · 2025-10-28

**Soundness:** 3
**Presentation:** 3
**Contribution:** 3
**Rating:** 6
**Confidence:** 3

**Summary:**

The paper introduces a model intrinsic measure dubbed local inconsistency, which can be used to predict generalization gap and regularize model training. This local inconsistency measure is grounded in information-geometry and is closely related to Fisher information matrix, Hessian matrix, and Gauss-Newton matrix. Moreover, the paper presents an algorithm to estimate local inconsistency empirically, based on which inconsistency-aware minimization (IAM) is proposed to train DNNs with improved generalization. Experimental results on image classification are provided under supervised, semi-supervised and self-supervised settings to validate the effectiveness of IAM.

**Strengths:**

1. The proposed measure has solid theoretic grounding.
2. The proposed measure can be estimated without labels.
3. The writing is clear and the proposed method is well positioned among related works.

**Weaknesses:**

1. In Table 1, results for both IAM-D and IAM-S are presented, where IAM-S gives better performance in general. In Table 2, only IAM-S is presented. However, for semi and self-supervised settings, only results for IAM-D are provided. What's the reason of this inconsistency? Is it because IAM-S works better in supervised learning while IAM-D works better in semi and self-supervised settings? If this is the case, then more analysis is needed to show why this happens and provide practical guidance on which variant of IAM should be preferred in which setting.
2. The number/type of model architectures used in experiments is quite limited. Experiments on more models, especially transformer-based models like ViT, are needed to show the generalizability of the proposed method.
3. How good is the convergence of Algorithm 1? In experiments, K is typically selected to be 1. Will more steps lead to better results? An ablation study on K could help.
4. The statement in Line 233-234 assumes there're only C nonzero eigenvalues. This assumption needs to be justified, for example, with some empirical results.
5. In Algorithm 2, is using the current minibatch good enough for computing $\delta_K$? Why not sample another class-balanced subset for computing $\delta_K$?
6. In Table 2, the performance of SGD with 400 epochs is worse than that of 200 epochs, which indicates overfitting. It's better to report results based on the best performing epoch (early stopping).
7. In Table 3, it's better to include FixMatch+SAM for comparison.
8. For self-supervised learning, it's better to provide linear probing results on CIFAR-100 as well.

**Questions:**

1. In Eq. 3, why is the constriant over $\delta$ based on $L_2$ norm but not $L_{\infty}$ norm which is more popular in adversarial training?

---

> ### Author Response · Authors · 2025-11-21
>
> > Inconsistency in reporting IAM-S vs. IAM-D
>
> We clarify the design philosophy and recommended usage of the two variants.
>
> #### IAM-D (Semi-/Self-Supervised):
> This variant adds an explicit regularization term ($S_\rho$) to the objective. Its key advantage is its **modularity and label-agnostic nature**. Since $S_\rho$ is computed solely from unlabeled data, IAM-D can be seamlessly "plugged in" to complex pipelines like FixMatch or SimCLR without significant engineering effort.
>
> #### IAM-S (Supervised):
> This variant implicitly minimizes local inconsistency by optimizing the loss at a perturbed point. Since it modifies the supervised loss gradient directly, it is highly effective for standard **supervised learning** (e.g., ImageNet). It is also more memory-efficient than IAM-D because it does not require storing separate gradients for the regularization term.
>
> Therefore, we recommend IAM-S for resource-constrained supervised tasks and IAM-D for scenarios where leveraging unlabeled data is the primary goal.
>
> ---
> >  The generalizability of IAM across model architectures (Evaluation on ViTs)
>
> To demonstrate the versatility of IAM beyond CNN architectures, we conducted additional fine-tuning experiments on ViT_s/16 pre-trained on ImageNet-1K using the CIFAR-10 dataset. We compared IAM-D against SGD and SAM.
>
> **Settings:**
> We fine-tuned the models for 10,000 steps with a batch size of 128, using SGD as the base optimizer. Gradient clipping with max_norm = 1.0 is applied. The initial learning rate was set to 0.01 with a linear decay schedule after 500 warm-up steps. For the perturbation magnitude $\rho$, we used $\rho=0.05$ for SAM and $\rho=0.1$ for IAM-D.
>
> **Table:** Test error ± stderr of SGD, SAM, and IAM-D when fine-tuning ViT-S/16 on CIFAR-10.
> |Method| |
> | --- | --- |
> | SGD | $1.86 \pm 0.01$ |
> | SAM | $1.56 \pm 0.01$ |
> | IAM-D | $1.52 \pm 0.02$ |
>
> As shown above, IAM-D is consistently competitive with SAM on a transformer-based architecture, confirming that our proposed local inconsistency measure is effective across different model inductive biases.
>
> ---
> > Convergence of Algorithm 1 (Ablation on K)
>
> We employed $K=1$ for IAM in our main experiments (Section 5) to ensure a fair comparison with SAM/ASAM in terms of computational efficiency (one additional backward pass per step).
> As described in Appendix E, $K=3$ is be sufficient to obtain a reasonable estimate of the maximizing direction.
> See Figure 6 in the revised paper (Appendix 3.1.1) for the convergence performance of Algorithm 1 versus the usual projected gradient ascent step.
>
> To investigate the effect of $K$, we conducted an ablation study using IAM-D on WRN28-10 (CIFAR-10/100).
>
> **Table**: Test error ± stderr and running time of IAM with respect to $K$.
> |$K$|CIFAR-10|CIFAR-100|runinng time (s/epoch)|
> |-|-|-|-|
> |1|3.28 ± 0.06|17.16 ± 0.03|239 (1.0x)|
> |2|3.03 ± 0.02|16.92 ± 0.04|311 (1.3x)|
> |3|2.99 ± 0.04|16.90 ± 0.03|378 (1.6x)|
> |5|2.98 ± 0.03|16.62 ± 0.02|525 (2.2x)|
>
> Unlike SAM, where more steps yield diminishing returns, IAM benefits from a more accurate estimation of local inconsistency as $K$ increases, offering a trade-off between performance and cost.
>
> ---
> > Mini-batch approximation in Algorithm 2
>
> Using the current mini-batch is a standard practice in stochastic optimization. While a single mini-batch may not be perfectly class-balanced, the stochastic nature of sampling ensures that most classes are covered over the course of an epoch. The iterative updates compensate for the noise of individual batches, allowing the model to approximate the geometry of the full data distribution effectively, without the I/O overhead of sampling a separate balanced subset.
>
> ---
> > ImageNet Results and overfitting at 400 Epochs
>
> First, we clarify that all reported results in our tables represent the best test error achieved during training.
> Regarding the 400-epoch SGD result: with a cosine learning rate schedule, simply extending the number of epochs does not guarantee better performance for standard SGD. In fact, stretching the schedule to 400 epochs caused SGD to overfit more compared to the 200-epoch schedule, resulting in worse "best epoch" performance (22.80% vs. 22.66%). Please See Figure 9 in the revised paper (Appendix D.2) for test error during training with a cosine learning rate schedule.
>
> However, **IAM-S mitigates this overfitting**. When trained for 400 epochs, IAM-S effectively utilizes the longer training horizon to improve generalization further, achieving **20.95%** Top-1 error. This contrast highlights IAM's ability to act as a robust regularizer that prevents overfitting even during prolonged training.

---

> ### Author Response · Authors · 2025-11-21
>
> > Comparison with FixMatch + SAM
>
> We did not originally include FixMatch + SAM because applying SAM to semi-supervised learning (SSL) is non-trivial and often unstable.
> #### 1. **Instability**
> Calculating perturbations based on pseudo-labels is noisy. If the model's prediction is incorrect, maximizing the loss against that target forces the model into an unstable state.
> #### 2. **Limited Scope**
> If we restrict SAM only to the *labeled* loss to avoid instability, we minimize sharpness only for the very small subset of labeled data (e.g., 250 samples). This fails to regularize the global landscape.
> To confirm this, we ran **FixMatch + SAM** on CIFAR-10 (250 labels).
>
> **Table**: Test error (mean ± stderr) on CIFAR-10 with 250 labels using a WRN-28-2 model.
> |Method|Test error (%)|
> |-|-|
> |FixMatch|6.26 ± 0.39|
> |FixMatch + SAM |9.90 ± 0.74|
> |FixMatch + IAM-D|**5.30** ± 0.08|
>
> This is significantly worse than FixMatch + SGD (6.26%) and FixMatch + IAM-D (5.30%). This failure case underscores the strength of IAM-D: it calculates inconsistency on **unlabeled data** without relying on potentially incorrect pseudo-labels, making it naturally superior for SSL.
>
> ---
> > SSL on CIFAR-100
>
> Our current SSL experiments on CIFAR-10 were intended as a proof of concept to demonstrate IAM's applicability to self-supervised learning. We are currently running the linear probing evaluation on CIFAR-100 and will include these results in the final revision to further validate the method's generalizability.
>
> ---
> > Q1. Why $L_2$ norm instead of $L_\infty$?
>
> Our choice of the $L_2$ norm is theoretically motivated by Information Geometry. Local inconsistency is derived from the second-order Taylor expansion of the KL divergence, which takes a quadratic form. Quadratic forms are naturally associated with ellipsoidal geometry and the Euclidean ($L_2$) norm. While $L_\infty$ is popular in adversarial training for pixel-space robustness, $L_2$ is more appropriate for capturing the curvature and sensitivity in the parameter space.

---

> > ### Comment · Reviewer_HS3y · 2025-11-27
> >
> > Thank the authors for their detailed response, which has addressed my major concerns. Thus, I will keep my positive score.

---

> ### Author Response · Authors · 2025-12-02
> **Further Response regarding theoretical assumption**
>
> > Theoretical/empirical justification for assumption ($\text{Tr}(F)/\lambda_{\max}(F) < C$).
>
> Justification for the Spectral Ratio Assumption in Deep Classification
> The geometric structure of the Hessian matrix in deep learning classification tasks has been rigorously established through both empirical observation and theoretical analysis.
> It is a widely accepted consensus that the Hessian spectrum is composed of a massive bulk centered near zero and a small number of dominant outliers.
> Sagun et al. (2018) provided early empirical evidence that the number of these large outliers corresponds to the number of classes, $C$.
>
> This phenomenon was mathematically formalized by Papyan (2018), who decomposed the Hessian into the Gauss-Newton matrix ($G$) and a residual term ($R$).
> Papyan (2018) demonstrated that the dominant outliers originate entirely from the $G$ term (which is asymptotically equivalent to the FIM, $F$), while the $R$ term contributes only to the bulk.
> Specifically, $G$ exhibits a hierarchical structure where the top $C$ eigenvalues are driven by the variance of class-wise gradients (Papyan, 2019;2020). Complementing this, Karakida et al. (2019) utilized mean field theory to prove that while the bulk eigenvalues remain small (order $O(1/M)$ or $O(1)$), the maximum eigenvalue $\lambda_{\max}$ scales linearly with the network width $M$, confirming the theoretical inevitability of huge outliers.
>
> Building upon this foundation, Karakida et al. (2021) theoretically analyzed the Fisher Information Matrix (FIM) for networks with Softmax outputs. They demonstrated that the inherent correlations introduced by the Softmax function (represented by the matrix $Q$) cause the top $C$ eigenvalues to disperse, unlike the degenerate outliers observed in linear output models. This dispersion, combined with the rank constraints of the Softmax function (effective rank $C-1$), provides a theoretical motivation for the assumption that $\text{Tr}(F)/\lambda_{\max}(F)$ is less than the number of classes $C$.
>
>
> These factors collectively suggest that the total variance (Trace) relative to the spectral norm ($\lambda_{\max}$) remains bounded below the number of classes $C$, which motivates the assumption $Tr(F)/λ_{\max}(F) < C$ in our analysis.
>
> ---
> **Reference**
>
> Levent Sagun, Utku Evci, V. Ugur Guney, Yann Dauphin, and Leon Bottou. Empirical analysis of
> the hessian of over-parametrized neural networks, 2018. URL https://arxiv.org/abs/1706.04454.
>
> Vardan Papyan. The full spectrum of deepnet hessians at scale: Dynamics with sgd training and
> sample size, 2018. URL https://arxiv.org/abs/1811.07062.
>
> Vardan Papyan. Measurements of three-level hierarchical structure in the outliers in the spectrum
> of deepnet hessians. In Kamalika Chaudhuri and Ruslan Salakhutdinov (eds.), Proceedings of
> the 36th International Conference on Machine Learning, volume 97 of Proceedings of Machine
> Learning Research, pp. 5012–5021. PMLR, 09–15 Jun 2019. URL https://proceedings.mlr.press/v97/papyan19a.html.
>
> Vardan Papyan. Traces of class/cross-class structure pervade deep learning spectra. Journal of
> Machine Learning Research, 21(252):1–64, 2020. URL http://jmlr.org/papers/v21/20-933.html.
>
> Ryo Karakida, Shotaro Akaho, and Shun-ichi Amari. Universal statistics of fisher information in
> deep neural networks: Mean field approach. In The 22nd International Conference on Artificial
> Intelligence and Statistics, pp. 1032–1041. PMLR, 2019.
>
> Ryo Karakida, Shotaro Akaho, and Shun-ichi Amari. Pathological spectra of the fisher information
> metric and its variants in deep neural networks. Neural Computation, 33(8):2274–2307, 07 2021.
> ISSN 0899-7667. doi: 10.1162/neco_a_01411. URL https://doi.org/10.1162/neco_a_01411.

---

### Official Review · Reviewer_9c4H · 2025-10-30

**Soundness:** 2
**Presentation:** 2
**Contribution:** 2
**Rating:** 4
**Confidence:** 4

**Summary:**

This paper introduces a new generalization measure called local inconsistency, which quantifies a model's output divergence with respect to parameter perturbations. Theoretically, the measure is shown to be governed by the Fisher Information Matrix, providing a robust, complementary signal to traditional sharpness. Based on this, the authors propose Inconsistency-Aware Minimization (IAM), an optimization framework that achieves generalization improvements comparable to SAM while showing performance improvement in unlabeled-data regimes like FixMatch and SimCLR.

**Strengths:**

1. The paper is clearly written and easy to follow.
2. Although the core idea is similar to SAM, it is still interesting to explore replacing the minimax loss with KL-divergence.
3. The paper provides some interesting theoretical insights.

**Weaknesses:**

1. Although the proposed method is a reasonable attempt at introducing a new measure, yet it gives little to no advantage over existing SAM and its variants.
2. I recommend that the authors consider incorporating FisherSAM [1], where $\rho$ is constrained within the KL region. This might lead to additional interesting results.

    [1] Fisher SAM: Information Geometry and Sharpness Aware Minimisation

3. Using gradient-based optimization to solve the inner maximization problem is valid and is similar to [2]. However, it could incur too much extra computation.

    [2] Regularizing neural networks via adversarial model perturbation

4. Based on my understanding and experience, directly optimizing KL-divergence may introduce difficulties, particularly due to its unbounded nature and potential numerical instability.

5. Does the method use the batch data in Algorithm 1? For supervised learning, I see no clear reason not using batch data.

**Questions:**

See weakness.

**Details Of Ethics Concerns:**

I have not found any discussions about the limitations and potential negative societal impact. But in my opinion, this may not be a problem, since the work only focuses on the optimization in deep learning. Still, it is highly encouraged to add corresponding discussions.

---

> ### Author Response · Authors · 2025-11-21
>
> > Advantages over SAM and its variant.
>
> We respectfully disagree and would like to clarify that IAM is *at least* competitive with SAM in standard supervised learning, while offering **clear and substantial advantages** in **semi-/self-supervised** regimes and in terms of theory and applicability.
> Reviewer 3aog mentioned:
> • Due to its label-agnostic nature, it can be effectively applied to semi-supervised learning and self-supervised learning. This is a clear advantage over methods like SAM which depend on a labeled loss.
>
> - **IAM offers a theoretically underpinned second-order metric regularization method with unlabeled data.**
> - Conceptually, IAM can be regarded as a label-agnostic, unlabeled-data extension of SAM, replacing loss-based sharpness with a local inconsistency regularizer.
>
> This is a core strength of local inconsistency and IAM, which we argue throughout the paper.
>
> #### In supervised learning, IAM is competitive with or better than SAM/ASAM
>
> Across CIFAR-10/100, F-MNIST, and SVHN, both IAM-D and IAM-S consistently reduce test error compared to SGD and are **on par with or better than SAM/ASAM** with the *same* backbone, data and training schedule. For example, on CIFAR-100, IAM-S improves over SAM by 0.81 percentage points in absolute error (16.82% vs 17.63%), and on SVHN, IAM-D/IAM-S improve over SAM by 0.34 percentage points (3.13% vs 3.47%).
>
> Our goal in the fully-supervised case is *not* to claim a dramatic gain over SAM, but to show that IAM is a **drop-in alternative that achieves SAM-level performance** while enabling additional capabilities below.
>
> #### Clear quantitative advantages in semi-supervised learning (where SAM is label-limited)
>
> IAM’s *label-agnostic* local inconsistency can be computed on both labeled and unlabeled samples within a minibatch, so the regularizer “sees” the full data distribution. In contrast, SAM can only use labeled points to define its objective, $\max_{\|\delta\| \leq \rho} \frac1B\sum_{i=1}^Bl(x_i, y_i;\theta+\delta)$.
>
> This leads to substantial performance gaps in label-scarce regimes where loss curvature can differ between the labeled set and the unlabeled set. Regularizing local inconsistency (IAM-D) on both labeled and unlabeled data shows a significant gap compared to SAM (Tables 3 and 4).
>
> Notably, SAM hardly improves generalization over SGD with very few labeled data on CIFAR-10 (250 labels). This may indicate that regularizing sharpness only on the labeled set does not effectively improve generalization for the overall dataset.
>
> #### Because local inconsistency is label-free, IAM can be added as a regularizer to the unlabeled objective.
>
> We show this by combining IAM-D with FixMatch:
>
> - CIFAR-10, FixMatch alone: 6.26% / 4.10% error (with 250 / 4000 labels)
> - FixMatch + IAM-D: **5.30% / 3.88%** error, improving a state-of-the-art SSL method without modifying its pipeline.
>
> SAM, by construction, depends on a labeled loss and does not naturally extend to these SSL scenarios; this is precisely the advantage already highlighted by another reviewer.
>
> To confirm this in FixMatch setting, we ran FixMatch + SAM (on the labeled loss) on CIFAR-10 (250 labels).
>
> **Table**: Test Error (mean ± stderr) on CIFAR-10 with 250 labels using a WRN-28-2 model.
> |Method|Test Error (%)|
> |-|-|
> |FixMatch|6.26 ± 0.39|
> |FixMatch + SAM  |9.90 ± 0.74|
> |FixMatch + IAM-D|**5.30** ± 0.08|
>
> This is significantly worse than FixMatch + SGD (6.26%) and FixMatch + IAM-D (5.30%).
> This failure case underscores the strength of IAM-D: it calculates inconsistency on unlabeled data without relying on potentially incorrect pseudo-labels, making it naturally superior for SSL.
>
> We agree that in fully-supervised benchmarks, IAM is **comparable** rather than uniformly superior to SAM or ASAM—this matches our own claims in the paper. However, the **combination** of:
>
> - strong supervised performance,
> - **large gains** in semi-/self-supervised settings, and
> - a new, label-free generalization measure with theoretical guarantees,
>
> constitutes a **meaningful advantage** over existing SAM-type methods, especially for practical scenarios where unlabeled data is abundant and labels are scarce.

---

> ### Author Response · Authors · 2025-11-21
>
> > Regarding recommendation about incorparating constraint within the KL region of FisherSAM.
>
> We think you may have misunderstanding on local inconsistency or FisherSAM.
>
> - Maximizing the objective within constraint on the (locally) same objective doesn’t lead to interesting results.
>
> FisherSAM has an idea to constrains the perturbation $\delta$ within KL, $\mathrm{KL}(f(\theta+\delta)\\|f(\theta)) \leq \rho^2$, which constraints distributional divergence between original parameter and perturbed parameter.
> Local inconsistency, $\max_{\|\delta\|\leq \rho} \mathrm{KL}(f(\theta)\\|f(\theta+\delta))$, measures the distributional divergence between original parameter and worst perturbed parameter, within an Euclidean ball of radius $\rho$ around the parameter $\theta$. This Euclidean ball constraint naturally connects between **local inconsistency** $S_{\rho}(\theta)$ and **top eigenvalue of Fisher information matrix** $\lambda_{\max} (F(\theta))$.
>
> If we change the Euclidean constraint of local inconsistency into KL constraint of FisherSAM,  $$\max_{\mathrm{KL}(f(\theta+\delta)\\|f(\theta)) \leq \rho^2} \mathrm{KL}(f(\theta)\\|f(\theta+\delta)) \quad \text{(?)}$$
> yields constant value $\rho^2$ with quadratic approximation, $\mathrm{KL}(f(\theta)\\|f(\theta+\delta)) \approx \tfrac12 \delta^\top F(\theta)\delta$ as we do, $\mathrm{KL}(f(\theta+\delta)\\|f(\theta)) \approx \tfrac12 \delta^\top F(\theta)\delta$ as FisherSAM does for small $\delta$. Further more, any perturbation $\delta$ satisfying $\tfrac12\delta^\top F(\theta)\delta = \rho^2$, can be the solution of the maximization problem.
>
> If you have another insightful and detailed opinions on incorporating FisherSAM, we are open to your valuable feedback.
>
>
> ---
> > Regarding the concern about extra computation.
>
> - IAM is computationally efficient since we use a single ascent step $K=1$ in practice.
>
> As described in section 5.1 and Appendix E of the paper (page 7 line 363, page 20 line 1030-1034), we use $K=1$ for Algorithm 1 (solving inner maximization problem) and Algorithm 1 require only  $K$ gradient computations per estimation. So **IAM**’s gradient-based optimization to solve the inner maximization problem incur **one additional backward pass** such as SAM and it’s variant (ASAM, FisherSAM, etc).
>
> ---
> > Based on my understanding and experience, directly optimizing KL-divergence may introduce difficulties, particularly due to its unbounded nature and potential numerical instability.
>
> Although the KL divergence $\mathrm{KL}(p \\| q)$ can in principle blow up when $p \approx 0$, in our metric we always evaluate it under the constraint $\|\delta\| \leq \rho$ with a small $\rho$. This local constraint makes the quantity much more stable than directly minimizing a standard KL objective such as $\mathrm{KL}(p \\| q)$, where the two distributions can be far apart. In addition, to further guard against potential numerical instabilities, we clamp the probabilities in $f(\theta)$ to a minimum value (e.g. $10^{−9}$) when computing the KL divergence. Please refer to the PyTorch implementation in the Supplementary Material for further implementation details.
>
> ---
> > Does the method use the batch data in Algorithm 1? For supervised learning, I see no clear reason not using batch data.
>
> As described in Algorithm 2, the perturbation $\delta^*$ is computed by Algorithm 1 using the current mini batch $\\{x_i\\}_{i=1}^b$ for supervised learning. For clarity, we add a comment about using mini batch. Please see Algorithm 2 and section 5.1 in page 7 of the paper.

---

### Official Review · Reviewer_3aog · 2025-10-31

**Soundness:** 4
**Presentation:** 3
**Contribution:** 3
**Rating:** 6
**Confidence:** 2

**Summary:**

This paper introduces "local inconsistency", a novel generalization measure for deep learning models that can be computed using only unlabeled data. This measure quantifies the model's output sensitivity to parameter perturbations. Based on this, the authors propose Inconsistency-Aware Minimization (IAM), an optimization method that incorporates this measure into the training objective. The paper demonstrates that IAM improves generalization in supervised learning (comparable to Sharpness-Aware Minimization, SAM) and, more importantly, enhances performance in semi-supervised and self-supervised learning scenarios where labeled data is scarce.

**Strengths:**

- The paper introduces $S_{\rho}$,  a novel measure of local output sensitivity. Its strength is that it can be computed from a single model using only unlabeled data.
- Due to its label-agnostic nature, it can be effectively applied to semi-supervised learning and self-supervised learning. This is a clear advantage over methods like SAM that depend on a labeled loss.
- Unlike other inconsistency measures that require training multiple models, "local inconsistency" can be computed from a single model.
- The proposed measure is theoretically grounded, with clear connections established to the Fisher Information Matrix (FIM) and the loss Hessian (Section 4.2), providing a solid information-geometric motivation.

**Weaknesses:**

- Similar to SAM, IAM is a min-max optimization algorithm. It requires an inner maximization step (Algorithm 1) to find the perturbation, which, even with K=1 step, effectively doubles the computational cost (requiring two gradient computations) per training step compared to standard SGD.
- For computational efficiency, the inner maximization step is approximated using only one step (K=1) of gradient ascent. While Appendix C argues this is meaningful, it is still a strong approximation and may not find the true worst-case perturbation.
- The method introduces new, sensitive hyperparameters, most notably $\rho$ and $\beta$. As shown in the appendix heatmap, these values are dataset-dependent and have a significant impact on performance.

**Questions:**

- The K=1 step for the inner maximization is justified for efficiency. Have you performed an ablation study on the number of steps K (e.g., comparing K=1 vs. K=3, as mentioned in Appendix E)? This would clarify if a more accurate estimation of the worst-case perturbation (K > 1) actually leads to better generalization, or if the K=1 approximation is sufficient.
- The hyperparameters $\rho$ and $\beta$ seem critical and dataset-dependent. Could the authors provide more intuition or a more principled guideline for setting these values beyond grid search?
- The paper proposes two variants: IAM-D and IAM-S. The results in Table 1 seem mixed (IAM-S is better on CIFAR-100, while IAM-D is slightly better on SVHN). Are there recommendations for when to use one versus the other? For instance, are there specific scenarios where IAM-D might be preferred over IAM-S?

minor
- (Typos) Page 17 and Page 19: pertubation -> perturbation

---

> ### Author Response · Authors · 2025-11-21
>
> > W1. Regarding the concern about computational cost for IAM
>
> - Like SAM and its variants, IAM requires one additional gradient computation compared to SGD.
> - IAM outperforms SGD under an equivalent computational budget.
> - we provide a solid theoretical foundation grounded in Information Geometry to explain how unlabeled data can improve model generalization.
> - IAM achieves **meaningful performance gains**, particularly in **label-limited settings**, by effectively extending regularization capabilities to unlabeled data.
>
> While IAM incurs additional computational overhead per step (similar to SAM), it outperforms SGD under an equivalent computational budget (i.e., allowing SGD to run twice as many epochs). The cost is further justified by IAM’s unique ability to extend regularization to unlabeled data using Information Geometry. Unlike SAM, which relies on labeled data, IAM leverages the Fisher Information Matrix (FIM) to regularize output sensitivity without explicit labels. This leads to significant gains in label-limited settings, which we believe outweighs the overhead. We also plan to explore computationally efficient variants in future work.
>
> ---
> > W2, Q1 Adequacy of $K=1$ and ablation study.
>
> - IAM only with $K=1$ still shows competitive performance with strong baseline.
> - It is standard practice to report results with one additional computation compared to SGD.
> - Ablation study on IAM with different values of $K$ shows differences compared to SAM.
>
> First of all, we also use $K > 1$ for estimating $S_ρ(θ)$ as metric for figure 1. As described in Appendix E, $K=3$ shows to be sufficient to get a reasonable estimate of the maximizing direction. See figure 6 in Appendix 3.1.1 of the revised paper for the convergence performance of Algorithm 1 versus the usual projected gradient ascent step.
>
> #### Rationale for $K=1$ and Comparison with SAM:
>
> Our motivation for employing $K=1$ in the paper was to match existing sharpness-aware minimization methods like SAM and ASAM in terms of computational efficiency. As detailed in our methodology, Algorithm 1 requires $K$ backward passes (gradient computations) for the inner maximization step.
> Standard SAM implementations typically utilize a single ascent step, incurring only one additional backward pass per update. Therefore, we set $K=1$ to match the computational budget of these baselines, ensuring that our improvements in generalization were not simply due to increased computations.
>
> Furthermore, prior work by Foret et al. (2021) reported that the test accuracy of SAM is not strongly sensitive to the number of inner maximization steps, with marginal gains observed from increasing iterations (e.g., 0.01–0.04 pp on CIFAR-10). Thus, reporting results with $K=1$ has been the standard practice for this class of optimizers.
>
> #### Ablation Study on $K$ (Question 1)
>
> Thank you for your suggestion.
>
> To address your question regarding whether a more accurate estimation of the worst-case perturbation ($K>1$) leads to better generalization, we conducted an ablation study using IAM-D on CIFAR-10 and CIFAR-100. We used $ρ=0.1$, $β=1$, and the standard hyperparameters described in Appendix E. The results are summarized below:
>
> **Table**: Test error and running time of IAM with respect to $K$ (WRN28-10).
> |K|CIFAR-10|CIFAR-100|running time (s/epoch)|
> |-|-|-|-|
> |1|3.28±0.06|17.16±0.03|239 (1.0x)|
> |2|3.03±0.02|16.92±0.04|311 (1.3x)|
> |3|2.99±0.04|16.90±0.03|378 (1.6x)|
> |5|2.98±0.03|16.62±0.03|525 (2.2x)|
>
> As shown in the table, we observe a consistent improvement in generalization performance as $K$ increases.
> This stands in contrast to SAM, where increasing the number of inner steps yields diminishing returns. While SAM saturates with a single step, our results indicate that for IAM, a more accurate estimation of the local inconsistency via multiple steps ($K>1$) provides tangible benefits to the final model performance.
>
> This suggests that the "local inconsistency" measure is robust and that better optimization of the inner objective directly translates to better generalization, offering a trade-off between efficiency and accuracy that users can tune based on their resources.
>
> We will include this ablation study and discussion in the revised paper to provide a more comprehensive view of the method's characteristics.

---

> ### Author Response · Authors · 2025-11-21
>
> ---
> > W3, Q2 Regarding the concern about sensitivity of the hyperparameter.
>
> - $β$ and $ρ$ are theoretically coupled as a single coefficient ($β ρ^2$), effectively reducing the hyperparameter degree of freedom to one.
> - $β=1$ consistently yields a competitive and stable baseline.
> - $ρ \in [0.1, 0.5]$ is a practical range for both IAM-S and IAM-D.
>
> #### Theoretical relation between $β$ and $ρ$
>
> As explained in Section 4.2, $S_ρ(θ)$ is approximated by the maximum eigenvalue of the FIM,
> scaled by $ρ^2$:
> $$S_ρ(θ) \approx \tfrac12 ρ^2 λ_{\max}$$
>
> Consequently, the penalty term in IAM-D, $β S_ρ(θ)$, can be reformulated as:
> $$β S_ρ(θ) \approx \frac{1}{2} (β ρ^2) λ_{\max}(F(θ))$$
>
> This reveals that the regularization strength is effectively governed by the single scalar coefficient $β ρ^2$. This implies that $β$ and $ρ$ are mechanistically coupled rather than acting as independent sources of sensitivity. In other words, a change in the neighborhood radius $ρ$ can be compensated by inversely scaling $β$ to maintain the same effective penalty. Therefore, despite the appearance of two hyperparameters, the effective degree of freedom for tuning the regularization strength is essentially one.
>
> **Sensitivity of $ρ$**
>
> Regarding the neighborhood size $ρ$, we found that values in the range of $[0.05, 0.5]$ robustly yield improved generalization with different settings and datasets. Please refer to the Figure 8 in Appendix D.1.2 the sensitivity of $ρ$. We acknowledge that $ρ$ is relatively sensitive compared to $β$. However, we would like to note that this sensitivity to the neighborhood radius is an inherent characteristic of geometry-aware optimization methods, including SAM and ASAM. Just as SAM requires tuning $ρ$ to define the appropriate scale of sharpness for a given topology, IAM requires tuning $ρ$ to define the appropriate scale for local inconsistency.
>
> **Practical Guidelines**
> Based on our empirical findings, we propose the following principled guideline for applying IAM:
>
> 1.  **Fix $β = 1.0$**: This value provides a balanced trade-off between the supervised loss and the inconsistency penalty across most datasets.
> 2. **Tune $ρ$ within $[0.05, 0.5]$**: Perform a minimal search for $ρ$ within this small range.
>
> This approach significantly reduces the tuning burden compared to a full grid search while still capturing the generalization benefits of the method.
>
> > Q3. Preferred scenarios with IAM-D or IAM-S.
>
> |Feature |IAM-S|IAM-D|
> |-|-|-|
> |Mechanism|Implicitly minimizes inconsistency by optimizing loss at a perturbed point (SAM-style).|Explicitly adds a regularization term ($S_ρ(θ)$) to the objective function.|
> |Primary Use Case| Standard supervised learning.|Semi-supervised & self-supervised learning.|
> |Key Advantages| Lower memory consumption.|Modular and label-agnostic; easily integrates into complex pipelines using unlabeled data.|
> |Data Focus|Optimizes based on labeled training samples.|Leverages unlabeled data to compute the inconsistency penalty.|
>
> We recommend choosing between IAM-S and IAM-D based on the specific learning scenario and resource constraints.
>
> #### 1. IAM-S for Supervised Learning
> IAM-S is designed as a "SAM-like" approach that implicitly minimizes local inconsistency by optimizing the loss at a perturbed point. Since it modifies the optimization trajectory of the supervised loss itself, it is inherently well-suited for **standard supervised learning** tasks. Furthermore, in terms of implementation, IAM-S generally exhibits **lower memory consumption** compared to IAM-D. Therefore, for supervised benchmarks like CIFAR-100 where memory and efficiency are priorities, IAM-S is often the preferred choice.
>
> #### 2. IAM-D for Semi- and Self-Supervised Learning
>
> On the other hand, IAM-D formulates local inconsistency as an explicit regularization term added to the objective. The key advantage of IAM-D is its **modularity and label-agnostic nature**. Because the regularization term $S_ρ(θ)$ can be computed solely from unlabeled data, IAM-D can be seamlessly integrated into complex pipelines—such as semi-supervised (e.g., FixMatch) or self-supervised frameworks (e.g., SimCLR)—without significant engineering effort. As demonstrated in our experiments (Tables 3 and Figure 3), IAM-D is the preferred variant for these scenarios where leveraging unlabeled data is the primary goal.

---

### Author Response · Authors · 2025-11-21
**Revision**

We uploaded Revision #1.

In the revised version, all red-colored text indicates modifications from the original version for improved clarity. We also added figures and tables to Appendix D, including the ablation study on $K$ (Table 4, Figures 6 and 7) and additional ViT experiments (Table 5), etc.

---

> ### Author Response · Authors · 2025-12-03
> **Revision 2**
>
> We uploaded Revision #2.
>
> In this revision, we clarified the meaningfulness of perturbation with $K=1$ step for the inner maximization and added ImageNet results with SAM, FixMatch results on CIFAR-100, discussion on failure case of FixMatch with SAM, etc.

---

### Author Response · Authors · 2025-12-03
**Summary Comment**

Dear AC and Reviewers,
We appreciate the reviewers for their constructive feedback.

## Strengths
We are encouraged that the reviewers consistently recognized the novelty and theoretical solidity of our proposed *local inconsistency* measure and the IAM framework.
Based on the reviews, we summarize the common strengths as follows:


### 1. Label-Free & Single-Model Computability (Acknowledged by 3aog, HS3y, aam4)
Reviewers highlighted the novelty and practicality of local inconsistency.
Unlike ensemble-based methods or standard SAM which require labeled loss, our measure can be computed using only unlabeled data and a single model.
- "Can be computed from a single model using only unlabeled data... a clear advantage over methods like SAM." (3aog)
- "The proposed measure can be estimated without labels." (HS3y)
- "Directly differentiable, enabling optimization within standard training pipelines." (aam4)

### 2. Solid Theoretical Foundations (Acknowledged by 3aog, 9c4H, HS3y, aam4)
All reviewers affirmed the theoretical rigor of our work, grounded in information geometry. They valued the established connections to the Fisher Information Matrix (FIM) and loss Hessian, as well as the generalization bound.
- "Theoretically grounded... providing a solid information-geometric motivation." (3aog)
- "Provides some interesting theoretical insights." (9c4H)
- "Solid theoretic grounding... closely related to FIM, Hessian." (HS3y)
- "Validate the single-model approximation theoretically (Theorem 4.1)." (aam4)

### 3. Versatility across Paradigms (Acknowledged by 3aog, aam4)
Reviewers confirmed the effectiveness of IAM beyond supervised learning, highlighting its seamless extension to semi-supervised and self-supervised learning (e.g., FixMatch, SimCLR) where labeled data is scarce. Notably, IAM significantly improves performance in label-scarce settings, a regime where SAM is often ineffective.
- "Effectively applied to semi-supervised and self-supervised learning." (3aog)
- "Shows consistent improvements across... semi-supervised... and self-supervised SimCLR." (aam4)

## Addressed Concerns

In this rebuttal, we have addressed all concerns by providing additional experiments (ViT, ablation on $K$) and theoretical clarifications. We summarize the key updates and responses below.
|**Concern**|**Reviewer's Point**|**Rebuttal**|
| :--- | :--- | :--- |
|**Advantages over SAM** (9c4H)| "it gives little to no advantage over existing SAM and its variants"|IAM offers **clear and substantial advantages in semi-/self-supervised regimes** and in terms of **theory and applicability**. Other reviewers agree with it. |
|**Computational Cost**| IAM could incur too much extra computation. | **The overhead is reasonable**. IAM (SAM and its variants) requires one more gradient computation than SGD, but with an equal compute budget, **IAM still outperforms SGD**.|
|**Verification on usage of K** (aam4)|Algorithm 1 use K=1 for efficiency, but the paper provides no empirical verification on why K=1 is enough.|We use $K=1$ to match the number of gradient computation in IAM and SAM. IAM ($K=1$) outperforms SGD, which validates using $K=1$ for efficiency just like SAM. Competitive performance of IAM-S suggests that **the perturbation direction of Algorithm 1 with $K=1$ is as meaningful as the gradient direction of SAM.** |
| **Justification for Assumptions** (aam4)| The paper didn't illustrate why the assumptions such as near-interpolation are reasonable. (Theorem 4.1)| Assuming the model **reaches a training loss of zero (near-interpolation)** is a **standard** (Zhang et al., 2017) and **well-accepted assumption** in deep learning theory (especially in over-parameterized regimes). As described in Theorem 4.1, other Assupmptions for Theorem 4.1 are inherited from Luo et al. (2024).|
|**Comparison with SAM in Table 2** (aam4)|ImageNet results compare only against SGD|As described in the paper, we clarify our original goal of ImageNet results is **to show the scalability of IAM**. We also show the result of SAM in ImageNet, which demonstrates that **IAM still competitive to SAM in Imagenet**.|

---

> ### Author Response · Authors · 2025-12-03
>
> ### 1. Difference from SAM (Response to Reviewer 9c4H and others)
> A primary concern was the distinction between IAM and SAM. We emphasize that IAM extends curvature regularization to be independent of the labeled loss, enabling its use on unlabeled data—a capability SAM lacks.
>
> - Crucial Evidence: We demonstrated that simply applying SAM in a semi-supervised setting (e.g., FixMatch) causes training instability and yields results worse than SGD. In contrast, IAM successfully regularizes curvature using unlabeled data, showing clear superiority in semi-supervised learning.
>
> - IAM-D vs. IAM-S: We clarified that IAM-S is suitable for supervised settings (memory-efficient, loss-dependent perturbation like SAM), while IAM-D provides a loss-independent regularization ideal for semi-/self-supervised learning. This structural difference is a key novelty.
> ### 2. Computational Cost & Efficiency (3aog, 9c4H, and aam4)
> - Fair Comparison: The computational burget of IAM ($K=1$) is aligned with the standard SAM (two gradient computations per step).
>
> - Performance Justification: Even when we **doubled the epochs for SGD** to match the computational budget, **IAM significantly outperformed SGD**, justifying the additional cost. We report the best score achieved by each SGD training run across either the standard epochs or the doubled epochs in revised paper.
>
>     **Table**: Test error ± stderr of SGD (400 epochs) and IAM (200 epochs, $K=1$).
>     |dataset|SGD|IAM-D|IAM-D (m=16)|IAM-S|IAM-S (m=16)|
>     |-|-|-|-|-|-|
>     |CIFAR-10 |3.68 ±0.04|3.28 ±0.06|3.03 ± 0.02|3.28 ±0.03|3.00 ±0.04|
>     |CIFAR-100|19.98 ±0.10|17.16 ±0.03|16.58 ± 0.05|16.82 ±0.01|16.53 ±0.10|
>
> - Ablation Study on $K$: As recomended, we provided an ablation study on the number of inner maximization steps ($K$) in Appedix D.1.1.
>
> ### 3. Theoretical Justification
> - Near-Interpolation Assumption (aam4): We clarified that assuming the model reaches a training loss of zero (near-interpolation) is a standard and well-accepted assumption in deep learning theory (especially in over-parameterized regimes) to analyze optimization behaviors.
>
> - Assumption on Spectrum of FIM (HS3y): We clarified that the sparse eigen spectrum of FIM are explored both theoretically and empirically in many previous works. Our assumption $\text{Tr}[F]/\lambda_{\max} (F) < C$ motivated by the fact that eigenvalues of FIM consist of dispersed and dominant $C$ outliers with and bulks near zero.
>
> ### 4. Clarifications on Misunderstandings (Reviewer 9c4H)
> - KL Divergence Constraint: The reviewer suggested incorporating a Fisher SAM constraint within a KL region. We respectfully clarify that maximizing KL divergence under a KL divergence constraint is mathematically contradictory in our context. Our objective is to find a perturbation that maximizes *local inconsistency* (measured by KL), distinct from the reviewer's suggested direction.
>
> - KL Optimization Stability: We addressed the concern about the unbounded nature of KL. As detailed in our supplementary material, our implementation ensures numerical stability, and the direction of our optimization (increasing *local inconsistency*) differs from standard model-fitting (minimizing KL to data).
>
> - Q. Does the method use the batch data in Algorithm 1? For supervised learning, I see no clear reason not using batch data.
> As described in original subbmitted paper, we clarified that it use mini-batch.
>
> ### 5. Additional Experiments
> - ViT Experiments: We have included fine-tuning results on Vision Transformers (ViT), demonstrating that IAM maintains its effectiveness across different architectures, performing better than SAM on CIFAR-10. We will add the fine-tuning results on CIFAR-100 in revised paper.

---

### Meta-Review · Area_Chair_bpX4 · 2026-01-06

**Summary:**

The four reviewers’ comments are mixed but identify several recurring concerns, including:
(1) the significant increase in computational complexity introduced by solving the inner maximization problem;
(2) the lack of justification for choosing 𝐾 = 1,  as well as the absence of an ablation study examining the impact of 𝐾;
(3) the sensitivity of the newly introduced hyperparameters 𝜌 and  𝛽, which appears to be dataset-dependent and has a substantial effect on performance;
(4) the lack of a clear advantage over Sharpness-Aware Minimization (SAM) and its variants;
(5) the limited scope of experimental evaluation, particularly with respect to the number and types of model architectures considered; and
(6) the lack of justification for the assumptions made in Theorem 4.1.

AC comments. Given the mixed nature of the reviews, the AC examined the paper at certain length. The AC finds the core idea presented in the paper to be interesting; however, the work is not yet sufficiently mature for the following reasons.

First, model generalization is fundamentally associated with discrepancies between training and testing data, i.e., variations in the model input space. In contrast, the paper primarily focuses on variations in the model parameter space of the model architecture. Although Theorem 4.1 provides an upper bound, the conceptual and theoretical connection between these two types of variations is not clearly established. Explicitly building and motivating this connection would significantly strengthen the paper, and the AC believes this is feasible.

Second, in light of the above, the limited experimental evaluation—particularly the narrow range of model architectures tested—becomes a notable limitation. A broader evaluation across diverse architectures is necessary to substantiate the generality and effectiveness of the proposed approach and would further strengthen the work.

**Reviewer Concerns:**

Most concerns were addressed to certain degree by the rebuttal. However, the concern on the limited experimental evaluation is still outstanding.

**Reviewer Scores:**

They would have likely maintained their scores.

---

### Decision · Program_Chairs · 2026-01-26

Reject